# Lexical Hints of Accuracy in LLM Reasoning Chains

## Abstract

Fine-tuning Large Language Models (LLMs) with reinforcement learning to produce an explicit Chain-of-Thought (CoT) before answering, produces models that consistently raise overall performance on code, math, and general-knowledge benchmarks. However, on benchmarks where LLMs currently achieve low accuracy, such as Humanity's Last Exam (HLE), they often report high self-confidence, reflecting poor calibration. Here, we test whether measurable properties of the CoT provide reliable signals of an LLM's internal confidence in its answers. We analyze three feature classes: (i) CoT length, (ii) intra-CoT sentiment volatility, and (iii) lexicographic hints, including hedging words. Using DeepSeek-R1, Claude 3.7 Sonnet, and Qwen-235B-Think on Humanity's Last Exam (HLE), a frontier benchmark with very low accuracy, Omni-MATH, a saturated benchmark of moderate difficulty, and GPQA-diamond, a graduate level scientific reasoning benchmark, we find that lexical markers of uncertainty (e.g., *guess*, *stuck*, *hard*) in the CoT are the strongest indicators of an incorrect response, while shifts in the CoT sentiment provide a weaker but complementary signal. CoT length is informative only on Omni-MATH and GPQA, where accuracy is already high ($\approx 70\%$), and carries no signal on the harder HLE ($\approx 9\%$), indicating that CoT length predicts correctness only in the intermediate-difficulty benchmarks, i.e., inside the model's demonstrated capability, but still below saturation. Finally, we find that uncertainty indicators in the CoT are consistently more salient than high-confidence markers, making errors easier to predict than correct responses. Our findings support a lightweight post-hoc calibration signal that complements unreliable self-reported probabilities and could support safer deployment of LLMs.

## 1 Introduction

Large Language Models (LLMs) can be fine-tuned with reinforcement learning to produce a Chain-of-Thought (CoT) before delivering their final response to improve their general performance (Jaech et al., 2024; Korbak et al., 2025; Muennighoff et al., 2025; Wei et al., 2022). The fine-tuning rewards are based both on the final correctness of the response—to improve capabilities—and on adherence to the CoT format, which may, for example, include safety-related specifications to enhance readability and alignment (Guan et al., 2024; Guo et al., 2025). Although LLMs have shown a strong performance increase on demanding benchmarks, such as GPQA (Rein et al., 2023), SWE-bench (Jimenez et al., 2024), and FrontierMath (Glazer et al., 2024), they often report very high confidence while still obtaining relatively low overall accuracy, reflecting poor calibration (Marjanović et al., 2025; Phan et al., 2025; Wei et al., 2024). This miscalibration masks silent failure modes and undermines the reliability of LLMs in open-ended settings (Chen et al., 2025a; Ke et al., 2025).

In addition to the poorly calibrated self-reported confidence (Phan et al., 2025), the confidence of LLMs in their final responses can be assessed via sampling agreement techniques that are computationally expensive (Lyu et al., 2025; Cherian et al., 2024; Kadavath et al., 2022), or by monitoring internal representations, which requires access to the model weights (Baek & Tegmark, 2025; Templeton et al., 2024; Zou et al., 2023). In contrast, the CoT of some LLMs is readily available with its final response, offering a potential proxy for assessing the model's internal confidence (Baker et al., 2025; Lindsey et al., 2025). Although it has been shown that an LLM's CoT may not always faithfully reflect the model's reasoning process and may be difficult

to interpret (Chen et al., 2025b; Kambhampati et al., 2025; Korbak et al., 2025; Stechly et al., 2025), there is some evidence that specific CoT properties predict the accuracy of the final response (Jiang et al., 2025; Wu et al., 2025). For example, there is mixed evidence on how informative the length of the CoT is for predicting response accuracy (Ballon et al., 2025; Chen et al., 2024; Shojaee et al., 2025; Su et al., 2025; Wang et al., 2025). Beyond quantitative properties such as length, recent studies have investigated how linguistic expressions of uncertainty relate to model correctness. For instance, Zhao et al. (2024) show that direct answers from LLMs tend to avoid epistemic markers altogether and, when they do appear, often overuse strengtheners—resulting in overconfident yet incorrect generations. More recent work has demonstrated that such linguistic cues can even be manipulated to affect model calibration (Zhou et al., 2023). In contrast, our analysis focuses on the chain-of-thought (CoT) of reasoning models, systematically quantifying naturally occurring lexical uncertainty markers across benchmarks and model families.

In this paper, we investigate whether Chain-of-Thought provides a reliable proxy for an LLM's internal confidence by systematically analysing three classes of features: (i) CoT length, aligning with prior work, (ii) intra-CoT sentiment volatility, and (iii) lexicographic indicators, such as hedging words. Whereas most existing studies focus narrowly on length-based correlations, our contribution is to extend this analysis with sentiment and lexical markers. We evaluate these features on three contrasting benchmarks: Omni-MATH, where state-of-the-art models already perform well; Humanity's Last Exam (HLE), which remains highly challenging; and GPQA-Diamond, a graduate-level reasoning benchmark designed to test conceptual depth and domain generalisation. The results show that CoT length and sentiment dynamics provide moderate predictive power in the "easier" benchmarks Omni-MATH and GPQA, but offer little signal on HLE. In contrast, lexical cues consistently yield informative calibration signals across all tasks.

## 2 Methods

### 2.1 Datasets and Evaluation

To analyse the lexicographic and linguistic characteristics of LLM reasoning, we collect the CoT and final responses from DeepSeek-R1 (Guo et al., 2025), Claude Sonnet 3.7 (Anthropic, 2025a), and Qwen-235B-Think (Alibaba Cloud, 2025) on three large-scale benchmarks of varying difficulty: Omni-MATH (Gao et al., 2024), GPQA (Rein et al., 2024), and HLE (Phan et al., 2025). We choose DeepSeek-R1, Claude 3.7 Sonnet (version: 20250219), and Qwen-235B-Think because their complete CoTs are fully available, and the serial test-time compute protocol that produces them—including the configurable maximum-token cap—is publicly documented (Guo et al., 2025; Anthropic, 2025b; Alibaba Cloud, 2025). Although the two models achieve comparable benchmark scores, Claude 3.7 Sonnet employs more extensive alignment techniques, such as constitutional fine-tuning (Bai et al., 2022), which may affect its calibration and the uncertainty signals visible in its CoT (Zhu et al., 2023).

Starting from the original $3,000$-question HLE dataset (Phan et al., 2025), we first remove 316 multimodal items. From the remaining $2,684$ questions, we then exclude 412 multiple-choice items, resulting in $2,088$ open-ended questions used in our analysis. The official benchmark results report that DeepSeek-R1, Claude Sonnet 3.7, and Qwen-235B-Think achieve an accuracy of 8.6% (calibration error: 81.4%), 8.9% (calibration error: 88.3%), and 11.8% (calibration error: 74.0%), respectively, on the full (text-only for DeepSeek-R1) version of the benchmark (for AI Safety, 2025). Omni-MATH comprises $4,428$ mathematics problems ranging from basic arithmetic to advanced algebra and geometry. On this benchmark, reasoning models, such as OpenAI o3-mini, achieve over 70% accuracy and even surpass 80% accuracy for algebra and calculus in a high reasoning effort setting (Ballon et al., 2025). Finally, we include GPQA-Diamond (Rein et al., 2024), a multiple-choice, graduate-level scientific reasoning benchmark covering physics, chemistry, and biology ($n = 198$ questions). GPQA-Diamond represents the most challenging tier of the GPQA suite, with official benchmark accuracies of 70.8% for DeepSeek-R1, 77.2% for Claude Sonnet 3.7, and 79.9% for Qwen-235B-Think. In contrast to the other models, Qwen-235B-Think exhibited an unusually high failure rate in producing valid answers in the required format. After a first pass (e.g., on Omnimath), fewer than 10% of the questions were returned with an interpretable answer, in sharp imbalance with the other models. To mitigate this, we implemented a rerun policy: each missing response from Qwen-235B-Think was resubmitted up to 10 times. Table 1 reflects this adjustment, where missing answers and confidence scores were filled

in under this rerun procedure. Even after this intervention, Qwen-235B-Think remains the model with the largest proportion of missing answers.

Omni-MATH, HLE, and GPQA-Diamond represent three complementary benchmarks that jointly span the reasoning spectrum. Omni-MATH offers 4,428 tiered mathematics problems with automatic, high-fidelity grading (Gao et al., 2024), allowing us to observe CoT behaviour across a continuous, well-calibrated difficulty spectrum. In contrast, HLE comprises 2,088 multidisciplinary, open-ended questions that push models to the limits of their reasoning abilities, often revealing brittle generalisation and poor calibration under genuine uncertainty (Phan et al., 2025). Finally, GPQA-Diamond (Rein et al., 2024) features 198 graduate-level, multiple-choice questions across physics, chemistry, and biology. While its performance levels are comparable to those on Omni-MATH, its broader scientific scope provides a useful test of cross-domain generalisation under similarly saturated model conditions. Note that the GPQA benchmark was run at a different time point, i.e. October 2025 versus February 2025.

For HLE and Omni-MATH, we prompt each model with a single, standardised template designed to elicit a clear explanation, a concise final answer, and a self-reported confidence score (0 %–100 %). We use a direct confidence elicitation format which closely follows the HLE benchmark (Phan et al., 2025). This verbalized confidence elicitation approach (Tian et al., 2023; Kadavath et al., 2022) captures models' self-stated belief in their correctness, providing a simple, interpretable, and model-agnostic measure of certainty. Alternative confidence proxies, such as token-level probability-of-correctness estimates (Kadavath et al., 2022), require logit access unavailable for closed-source models (e.g., Claude 3.7 Sonnet) and were thus not considered here.

> Your response should be in the following format:
>
> Explanation: your explanation for your final answer
>
> Exact Answer: your succinct, final answer
>
> Confidence: your confidence score between 0 % and 100 % for your answer
>
> Solve the following problem:

For GPQA-Diamond, we employed a slightly adapted version of the prompt provided by the Epoch AI benchmark suite (AI, 2024) to align with its multiple-choice format and encourage well calibrated responses (Kadavath et al., 2022).

> The last line of your response should be of the following format:
>
> 'Answer: LETTER, Confidence: your confidence score between 0% and 100% for your answer' (without quotes) where LETTER is one of ABCD.
>
> Answer the following multiple choice question.
>
> {Question}
>
> A) {Option A}
> B) {Option B}
> C) {Option C}
> D) {Option D}

Additionally, for Claude Sonnet 3.7, we set its max tokens to 128k and its thinking budget to 100k tokens. Similarly, for Qwen-253B-Think, we use the model presets of 131.1k max tokens, 32.2k of which can be used for output. DeepSeek-R1 has a max tokens parameter set to 128k, which includes the CoT tokens.

To grade the final responses, we employ a customised grading strategy that accommodates the unique challenges and features of each benchmark. Omni-MATH's responses were automatically scored using Omni-Judge (Gao et al., 2024), an automatic mathematical grader, whose alignment with human judgment has been explicitly validated for Omni-MATH (Ballon et al., 2025; Gao et al., 2024; Verga et al., 2024).

We prompted Omni-Judge as follows:

---

**OBJECTIVE**

You are tasked with evaluating the correctness of a student's answer. Below, you are provided with a problem, a reference answer, and a student's answer. You should assess whether the student's answer captures the same meaning as the reference answer, even when expressed with different wording or format.

Your tasks include:
A. Identify Mathematical or Notational Equivalence.
B. Conclude with a brief explanation as to why the student's output is correct or incorrect.

**# RESPONSE: MARKDOWN REPORT #**
**Student Final Answer**
Extract the student's final answer, which is enclosed in "\boxed{}".

**Equivalence Judgement**
Whether the student's answer shares the same meaning with the reference answer. (TRUE or FALSE)

**Justification**
Conclude with a brief explanation as to why the student's answer is correct or incorrect.

**ATTENTION**
- The reference answer is ALWAYS correct. You should carefully judge whether the student gives the same answer as the reference answer.
- The answer is FALSE even if the student's final answer is almost correct with a minor mistake.
- The answer is contained within the "boxed" section, so you can focus solely on comparing the content in the student's answer box with the reference answer, without needing to consider the intermediate steps.
- Add "=== report over ===" at the end of the report.

---

For HLE, each answer from DeepSeek-R1 and Claude 3.7 Sonnet is independently assessed by both a human expert and a different LLM. We use OpenAI o3-mini to avoid preference leakage, which can occur when the same model is used for both answer generation and evaluation, potentially resulting in more lenient or biased assessments of its own responses (Li et al., 2025). The agreement between human and automated grades is high: 93.9% for Claude 3.7 Sonnet and 93.4% for DeepSeek-R1, corresponding to Cohen's kappa values of 0.88 and 0.87, indicating near-perfect alignment. For further analysis, we include only HLE items where human and OpenAI o3-mini evaluations agree. For Qwen-235B-Think, grading was performed exclusively using OpenAI's newer o4-mini, which provides improved calibration and evaluation robustness relative to the o3-mini-judge used on the answers of the other two reasoning models.

Finally, to assess model calibration, we compute the expected calibration error (ECE), by grouping the outputs into confidence bins and calculate the mean absolute difference between the average reported confidence of each bin and its empirical precision.

Across the three benchmarks, DeepSeek-R1, Claude Sonnet 3.7, and Qwen-235B-Think perform broadly in line with their official benchmark results, showing no single model as a consistent winner in accuracy. Note that a small derivation is to be expected as (Bowyer et al., 2025) argue, point estimates on small, specialised tests should be reported with confidence intervals to avoid underestimating uncertainty (e.g., error bars or

significance tests). On HLE, Qwen achieves the highest accuracy (13.5%), followed by Claude (9.2%) and DeepSeek-R1 (8.6%). On Omni-MATH, DeepSeek-R1 leads (72.5%) ahead of Claude (69.1%) and Qwen (60.6%), while on GPQA, Qwenn attains the highest accuracy (77.8%), closely followed by Claude (75.8%). On GPQA DeepSeek-R1 is a distant third scoring (61.6%), 9% below their official score. Calibration patterns, however, reveal a more systematic trend: all models exhibit strong confidence in their predictions, with most responses clustered near maximal self-reported certainty (Appendix B). Among them, Qwen-235B-Think achieves the lowest ECE across all benchmarks, however this pattern is not reflected in alternative measures of calibration such as the Macro Callibration Error (MacroCE) (Lin et al., 2022; Si et al., 2022; Zhao et al., 2024) and Brier score (Brier, 1950).

A global overview of accuracies and calibration errors is presented in Table 1. A full breakdown, categorised by difficulty level (only for Omni-MATH), is provided in Appendix A.

Table 1: Overall accuracy and calibration error for DeepSeek-R1, Claude 3.7 Sonnet, and Qwen-235B-Think across the HLE, Omni-MATH, and GPQA benchmarks. While Claude achieves the highest accuracy on GPQA, DeepSeek-R1 performs best on Omni-MATH, and Qwen leads on HLE. Calibration error remains highest for Claude across all benchmarks, suggesting overconfidence, while Qwen shows markedly better calibration on Omni-MATH but with a substantial number of missing confidence scores (1,631 omitted cases). Expected Calibration Error (ECE) is computed, as in (Guo et al., 2017), by computing the weighted mean absolute difference between predicted confidence and empirical accuracy across binned confidence levels reported by the model. We also report Macro Calibration Error (MacroCE) or Mean Absolute Deviation (Lin et al., 2022; Si et al., 2022; Zhao et al., 2024), the unweighted mean absolute calibration gap across non-empty bins, and the Brier score (Brier, 1950), the mean squared error between predicted probabilities (self-reported confidence) and binary outcomes.

| Dataset | Model | Accuracy (%) | ECE (%) | MacroCE (%) | Brier Score (%) | Missing Answer (%) | Missing Confidence Score (%) |
|---|---|---|---|---|---|---|---|
| HLE | DeepSeek-R1 (Jan) | 8.6% | 78.3% | 49.0% | 70.8% | 0.0% | 0.0% |
| | Claude 3.7 Sonnet | 9.2% | 82.9% | 55.5% | 78.8% | 0.0% | 0.0% |
| | Qwen-235B-Think | 13.5% | 78.0% | 57.7% | 74.9% | 20.0% | 20.0% |
| Omni-MATH | DeepSeek-R1 (Jan) | 72.5% | 22.2% | 32.1% | 22.1% | 0.0% | 9.5% |
| | Claude 3.7 Sonnet | 69.1% | 29.1% | 47.2% | 29.0% | 0.0% | 2.0% |
| | Qwen-235B-Think | 60.6% | 8.8% | 45.8% | 10.1% | 31.6% | 36.8% |
| GPQA | DeepSeek-R1 (Jan) | 61.6% | 29.3% | 41.8% | 27.9% | 0.0% | 0.0% |
| | Claude 3.7 Sonnet | 75.8% | 17.5% | 31.2% | 20.4% | 0.0% | 0.0% |
| | Qwen-235B-Think | 77.8% | 16.1% | 40.8% | 18.9% | 1.0% | 1.0% |

## 2.2 Intra-CoT sentiment Volatility in the Chain-of-Thought

The ability to regulate emotions, such as maintaining composure or overcoming moments of doubt, has been linked to improved cognitive outcomes in humans (Schutz et al., 2004; Schutz & Davis, 2000; Schutz et al., 2008; Harley et al., 2021). Motivated by this, we examine the impact of the Intra-CoT sentiment volatility within the CoT responses on the response correctness. Specifically, we asked whether the model's reasoning began with a confident, optimistic tone and ended in doubt, or if it started hesitantly and grew more optimistic as the answer developed.

To quantify this, we process each CoT through OpenAI o3-mini as an independent sentiment evaluator (Tori et al., 2024). For each rationale, we extract two sentiment scores—the most negative and the most positive sentiment, ranging from -1 (strongly negative) to +1 (strongly positive). We also record the order in which these sentiments appear in the CoT, i.e., whether the response moved from negative to positive (an "upward" direction, scored as 1), from positive to negative ("downward," scored as –1), or remained neutral throughout (scored as 0). We use the following prompt to evaluate each CoT:

> Judge how emotional this text is by reporting on its most positive emotion and its most negative emotion on a scale from -1 (negative) to 1 (positive), and the order in which they appear, i.e. direction in which they appear, if the most negative emotion appears before the most positive emotion the direction is upward so, 1, if inverse is true the direction is -1 if the text is completely neutral and no gradient exists present between the two emotional extremes the direction is 0, picking up on small effects. Always end your response with:
>
> Most positive emotion Score: ...
> Most negative emotion Score: ...
> Direction: ...

Using the sentiment scores provided by OpenAI o3-mini, we define Intra-CoT sentiment volatility as the signed difference between the most positive and most negative sentiment within each CoT. Large differences ($\Delta$) may indicate moments of uncertainty, backtracking, or sudden shifts in confidence, while smaller deltas suggest a more consistent and steady tone throughout the reasoning process.

### 2.3 Lexicographic Analysis of the Chain-of-Thought

To explore the connection between the linguistic characteristics of model-generated rationales and their accuracy, we perform a lexicographic analysis of CoT responses from all evaluated datasets and models. For each unique word found within a CoT, we calculate its relative accuracy by comparing the success rate of responses containing that word to the average accuracy within the respective dataset. To ensure reliability, we exclude low-frequency words by requiring that each word appear at least a minimum number of times (300 occurrences) in every dataset to be included in the analysis.

#### 2.3.1 Hedging Words and Accuracy

In addition to the general lexicographic analysis, we focus on a specific subset of the lexicon: hedging words. In human communication, hedging—through modal verbs (e.g., might, could), uncertainty adverbs (e.g., possibly, perhaps), and tentative phrases (e.g., it seems that)—serves as a cue indicating lower confidence (Hyland, 1998; Demir, 2018; Lakoff, 1973). We hypothesise that the same linguistic uncertainty cues may also appear in the CoT of reasoning models. Prior work has shown that non-reasoning LLMs tend to underuse epistemic markers, often defaulting to overconfident phrasing even when incorrect (Zhou et al., 2024), motivating our investigation of whether hedging functions differently in explicit reasoning traces.

Drawing on established taxonomies of uncertainty markers, (Hyland, 1998; Demir, 2018; Lakoff, 1973), we compiled a lexicon of such expressions and computed a hedging rate per, i.e., proportion of sentences in a rationale containing at least one hedging expression CoT. The full hedging lexicon used was:

- Modal and uncertainty verbs: might, may, could, should, would, seems, suggests, appears

- Uncertainty adverbs: possibly, perhaps, likely, unlikely, probably, generally, usually, sometimes, often, tends, somewhat, rather, quite, almost, nearly, virtually, presumably, arguably, relatively, fairly, reasonably, mostly, partially, mainly, primarily, essentially, basically

- Common hedging phrases: it seems that, it appears that, it suggests that, it is possible that, it is likely that

- Additional qualifiers: in part, to some extent

## 3 Results

### 3.1 Relationship Between CoT Length and Accuracy

To assess how the length of the CoT relates to response correctness, we analyze accuracy as a function of the number of words in each CoT rationale. We observe a downward trend in the Omni-MATH and GPQA

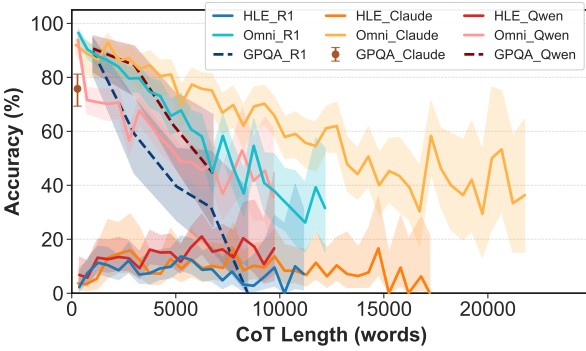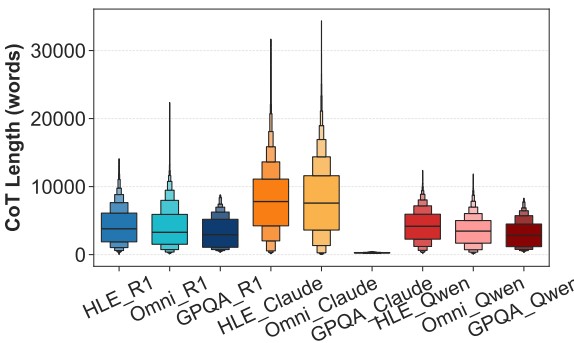

Figure 1: Accuracy as a function of CoT length (top) and the distribution of CoT lengths (bottom) across all benchmarks and model families. Across Omni-MATH and GPQA, accuracy decreases with increasing chain length for all three models. On HLE, accuracy remains uniformly low regardless of length. Claude consistently produces the longest and most variable reasoning chains, whereas Qwen tends to generate the shortest reasoning chains. CoT length bins with fewer than 30 samples were excluded from the accuracy plot. The GPQA–Claude results appear as a single point with fences, as Claude's reasoning chains were uncharacteristically short for this benchmark. We speculate this might be due to a platform update on Anthropic's side between benchmark runs.

benchmarks: accuracy declines as CoT length increases (see Figure 1). Claude 3.7 Sonnet proves more robust to longer reasoning chains than DeepSeek-R1 and Qwen-235B, losing on average 3% of accuracy per additional 1,000 CoT words, compared to 4.4% for Qwen-235B and 5.8% for DeepSeek-R1. These results echo the findings of (Ballon et al., 2025), who report that o3-mini-high loses about 0.8 % accuracy per 1,000 additional reasoning tokens on Omni-MATH benchmark. A similar pattern extists for GPQA, where DeepSeek-R1 and Qwen-235B lose 11.1% and 8.4% accuracy per 1000 additional CoT words. We do not report this trend for Claude-3.7 Sonnet, as all of reasoning chains were uncharacteristically short, all less than 1000 words, for the GPQA benchmark. This could be a consequence of the later date on which GPQA was run. In contrast to the downward pattern reproted for Omni-MATH and GPQA, in HLE, we find that the CoT length shows no discernible pattern, accuracy stays low, and fluctuates widely regardless of CoT length. This suggests that CoT length predicts correctness only for benchmarks of intermediate difficulty—that is, tasks within the model's demonstrated capabilities but not yet saturated. These findings are consistent with previous work showing a negative relationship between CoT length and response accuracy on Omni-MATH (Ballon et al., 2025), as well as the disappearance of this effect on the most challenging benchmarks (Shojaee et al., 2025; Opus & Lawsen, 2025). In general, we find that Claude Sonnet 3.7 generally produces longer CoTs than DeepSeek-R1 and Qwen-235B, which may be due to architectural choices or differences in reinforcement learning post-training (Muennighoff et al., 2025; Guo et al., 2025). In contrast, Claude Sonnet 3.7 produced the shortest CoT on GPQA, which may reflect the timing of the evaluation conducted in October rather than February, when the model was closer to its release.

### 3.2 Intra-CoT sentiment Volatility and Reasoning Dynamics

Figure 2 shows the relationship between Intra-CoT sentiment volatility within CoT and response correctness. We observe that a predominantly parabolic pattern emerges for Omni-MATH, with accuracy peaking at a volatility value of approximately 0.1. This suggests that CoTs with a consistent or slightly uplifting sentiment are associated with the highest accuracy. The answers asociated with GPQA, tend to be more correct as a model's exhibits neutral or less downward sloping reasoning. In contrast, no such pattern is observed for HLE, where accuracy remains uniformly low regardless of the degree of sentimental variation. To rule out the possibility that a nonzero mean (baseline) emotion score was driving this effect, we repeated the volatility-accuracy analysis on mean-centred sentiment values. After centring, we found the same parabolic, upward trend for Omni-MATH and GPQA, and the same flat pattern for HLE. While all three reasoning

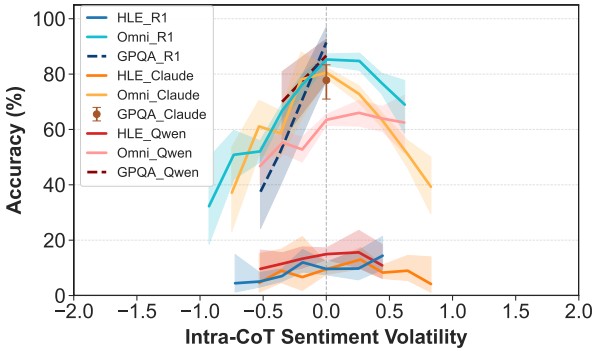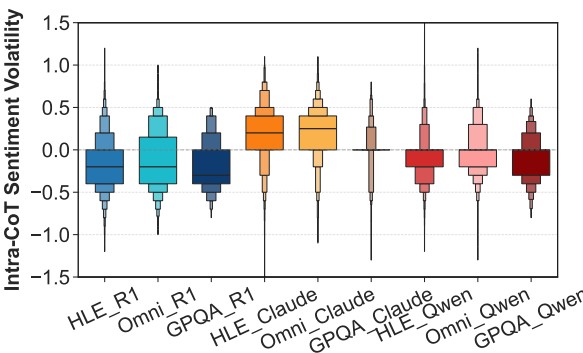

Figure 2: Accuracy as a function of intra-CoT sentiment volatility (left) and its distribution (right) across all benchmarks and model families. On Omni-MATH and GPQA, accuracy peaks for CoTs with low or slightly positive volatility, suggesting that mild emotional fluctuation within the reasoning process supports better outcomes. In contrast, HLE shows uniformly lower accuracy across all volatility levels. Claude 3.7 Sonnet tends toward positive sentiment, while DeepSeek-R1 and Qwen-235B lean toward mildly negative reasoning chains. For GPQA–Claude, only a single averaged point with fences is shown due to the limited sample size at each volatility level. Bins with fewer than 30 samples were excluded.

models display a predominantly neutral CoT, they exhibit markedly different sentimental footprints. Claude 3.7 Sonnet's responses tend to be slightly optimistic in tone, becoming more positive over the course of their reasoning, while DeepSeek's and Qwen rationales often exhibit a slight negative shift, see Figure 2(right).

### 3.3 Lexical Hints of Uncertainty

To scrutinise how specific words reflect or undermine a model's underlying confidence in response correctness, we construct a lemmatised lexicon from the CoT. We exclude rare or highly exclusive words from the lemmatisation process, retaining only those appearing in at least 300 CoT responses for each model and benchmark. We distinguish between lemmatised words that consistently harm or boost accuracy. For ease of reading, we use words to refer to the lemmas in the remainder of the paper.

Interestingly, many of the words most strongly linked to reduced accuracy, such as *complicated, complex, likely, probably, often, unless*, are intuitively associated with task difficulty and cognitive overload, mirroring language that humans use to express uncertainty or task difficulty (Renkl, 1997; Hyland, 1998; Demir, 2018; Lakoff, 1973). This suggests a degree of linguistic convergence in the signalling of uncertainty between reasoning models and humans. As shown in Figure 3, the top 25 words most detrimental to accuracy also include terms conveying confusion or lack of direction, such as *depend, unless, else, direction*. A full overview of these lexical hints of uncertainty, i.e., "harmful" words can be found in Appendix 4.

In contrast, only the word "equation" appears to have consistently boosted accuracy across all datasets (see Figure C). However, when considering its confidence interval, this apparent effect is likely due to random variation rather than a true underlying pattern. As a general rule, the words with the highest mean relative accuracy tend to be those characteristic of mathematical reasoning, including *equivalent, solve, therefore, equal, equation, calculation, formula, compute*, and *coefficient*. This is an artefact of Claude, DeepSeek, and Qwen demonstrating above-average performance on mathematical questions within HLE, achieving accuracies of 10%, 9.1%, and 13%, respectively.

#### 3.3.1 Hedging

We continue by looking at a subset of our full lexicon, hedging words. Hedging words are typically used by humans to mark doubt or a lack of knowledge (Hyland, 1998; Demir, 2018; Lakoff, 1973); the same could be true for LLM-generated CoT.

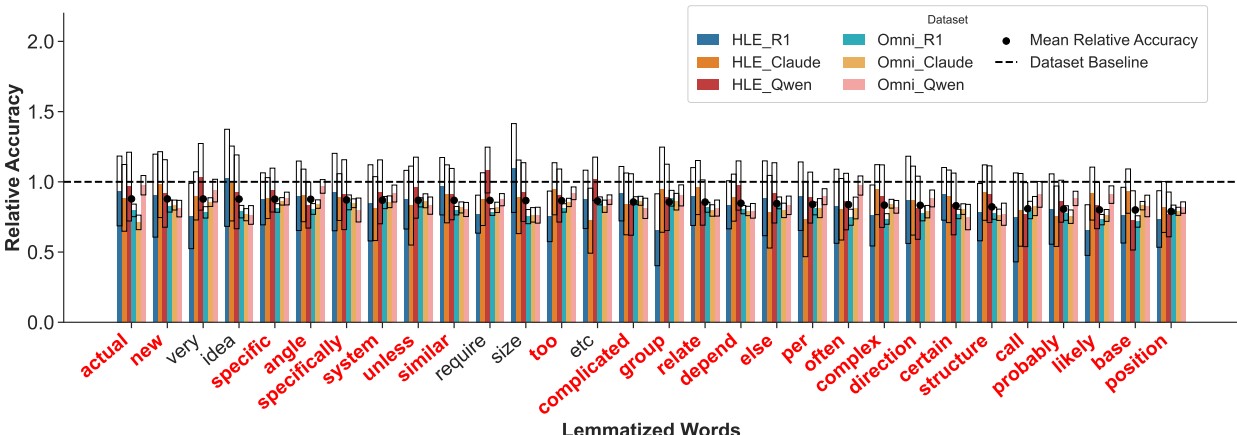

Figure 3: Relative accuracy of the 30 most detrimental lemmatised words across datasets and models. t Many of the most harmful words (e.g., complicated, complex, likely, probably, depend, often, unless) express uncertainty, cognitive overload, or confusion. The GPQA benchmark was excluded from this analysis because its limited size prevents the detection of robust lexical signals (i.e., words appearing $\geq 300$ times). When the occurrence threshold is lowered (e.g., $n = 50$), the benchmark becomes noisy and sparse (only outputting 45 words), with spurious signals driven by words that appear verbatim in the question text (e.g., group, structure, position, product).

Across the eight of the nine model benchmark pairs, we observe a negative relationship between hedging rate and accuracy: as the proportion of hedged sentences increases, the accuracy tends to decline (Figure 4). This trend is most pronounced for Omni-MATH, where the correlations are $r = -0.24$ ($p < 0.001$) for DeepSeek-R1 and $r = -0.14$ ($p < 0.001$) for Claude 3.7 Sonnet, indicating small but statistically detectable negative associations under the null hypothesis of zero correlation. In contrast, Qwen-235B-Think exhibits a weak positive correlation ($r = 0.06$, $p < 0.001$), a possible effect from the large amount of unanswered questions. For HLE, the correlations are weaker: $r = -0.10$ ($p < 0.001$) for DeepSeek-R1, $r = -0.04$ ($p = 0.06$) for Claude 3.7 Sonnet, and $r = -0.07$ ($p < 0.01$) for Qwen-235B-Think. All three exhibit small negative effects, though only DeepSeek and Qwen reach statistical significance. The hedging relationship is the most pronounced for GPQA, where correlations reach $r = -0.38$ ($p < 0.001$) for DeepSeek-R1, $r = -0.21$ ($p < 0.01$) for Claude 3.7 Sonnet, and $r = -0.25$ ($p < 0.001$) for Qwen-235B. Across all benchmarks, higher hedging rates are generally linked to lower accuracy—most clearly in Omni-MATH and GPQA—though the effect weakens on HLE. Median hedging rates were about 9% on Omni-MATH, 18% on HLE, and 22% on GPQA.

## 3.4 Prediction Accuracy

The question now remains whether these lexicographic features can be used to predict/anticipate the correctness of a reasoning model's final response. To assess which features best predict the correctness of model-generated reasoning, we train a neural network on both general features (CoT length, sentiment volatility, and hedging rate) and with the 25 most consistently "harmful" non-lemmatised words (see Appendix 3, and combinations between the standard features and "harmful" words. To prevent lexical leakage, the harmful-word lexicon is rebuilt for each of the 30 random seeds using only the training data. Each model is trained on one benchmark and evaluated both within and across benchmarks to test in- and cross-benchmark predictive capability. The GPQA benchmark is used exclusively as a held-out test set, given its smaller size, which prevents consistent identification of lexical cues, (e.g., words appearing at least 300 times), which is especially true for the uncharacteristically short reasoning chains of Claude Sonnet 3.7 on GPQA.

For all experiments, we employ a feed-forward neural network with two hidden layers (32 and 16 units, ReLU activations) and a sigmoid output. Models are trained using the Adam optimiser and binary cross-entropy loss. Given the strong class imbalance in the training data, particularly for HLE, whose accuracy for the

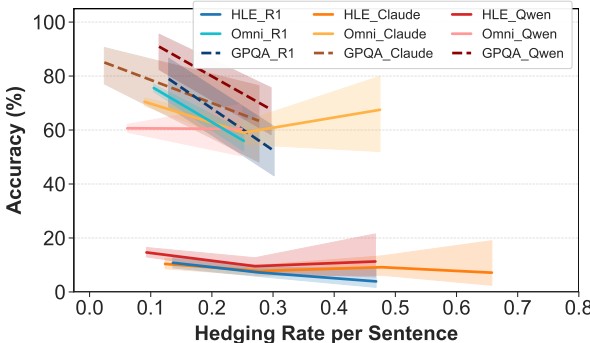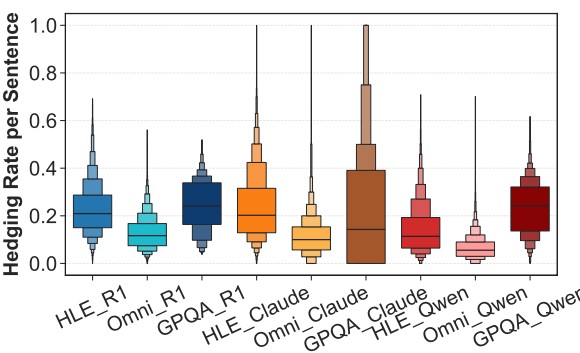

Figure 4: Accuracy as a function of hedging rate (left) and the distribution of hedging rates (right) for each model and benchmark. Higher hedging rates are generally associated with lower accuracy, particularly on Omni-MATH and GPQA. The effect is weaker for HLE and nearly absent for Claude 3.7 Sonnet. Hedging is most common in GPQA (median $\approx 22\%$) and HLE (median $\approx 18\%$), while low for Omni-MATH (median $\approx 9\%$). As before, bins with fewer than 30 samples were excluded from the left figure.

minority class was only 8.9%, it is crucial to prevent the model from exploiting this imbalance with trivial predictions (e.g., always guessing "incorrect" for DeepSeek's answers on HLE would yield an accuracy of 91.4%). Consequently, we apply class weighting during neural network training, penalising mistakes on the minority class more heavily, to encourage the model to learn non-trivial, meaningful patterns for both classes. All experiments are conducted on 30 seeds (0-29) to ensure robust estimation, with the results averaged across multiple runs. Training is performed on 80% of the original unbalanced data; we then balance the remaining 20% so that evaluation sets contain the same number of correct and incorrect examples, ensuring that performance consistently above 50% is indeed better than random guessing.

Figure 5 shows that lexical features alone are the strongest individual predictors of correctness, generalising well both within and across benchmarks. Adding additional features, such as CoT length, hedging rate, or sentiment dynamics, does not improve predictive accuracy and, in some cases, even decreases it. The only exception is the combination of lexical features with self-reported confidence, which yields the highest mean accuracy across all experimental settings. Notably, CoT length, which has been highlighted in previous studies (Ballon et al., 2025; Chen et al., 2024; Shojaee et al., 2025; Su et al., 2025; Wang et al., 2025), consistently provides the least predictive value and harms performance when included as an additional feature in all of our analyses. The ROC curves for each analysis can be found in Appendix D.

To assess whether the lexical signal merely captures domain composition (i.e., non math-related terminology) or interacts with reasoning length, we conducted an auxiliary domain-sensitivity analysis that split the HLE test set into mathematical and non-mathematical questions while jointly controlling for CoT length, see Table 5. The lexical-only models performed worse on math than on non-math items (0.52 vs. 0.56 accuracy). When reasoning length was added as a covariate, this difference disappeared—but overall accuracy declined—indicating that CoT length partly mediates domain variance without providing additional discriminative power. Adding confidence to the lexical markers also levels performances over mathematical and non-mathematical questions, but at the same time improves accuracy to aproximately 0.58. These results support the view that CoT length acts as a coarse marker of reasoning effort that is most informative for intermediate-difficulty regimes (e.g., Omni-MATH, GPQA). Furthermore, they reflect what we find in the neural networks, who show that the models trained on the reasoning length of Omni-Math perform well within omni-math and GPQA but do not generalize to the harder HLE.

Overall our findings suggest that a small set of words offers an important and potentially generalizable signal for predicting final response correctness, serving as lexical hints of uncertainty within CoT reasoning. The complete confusion matrices and Matthews Correlation Coefficient (MCC) for the best-performing neural network, which was trained using only these 25 words, are available in Appendix Table 6.

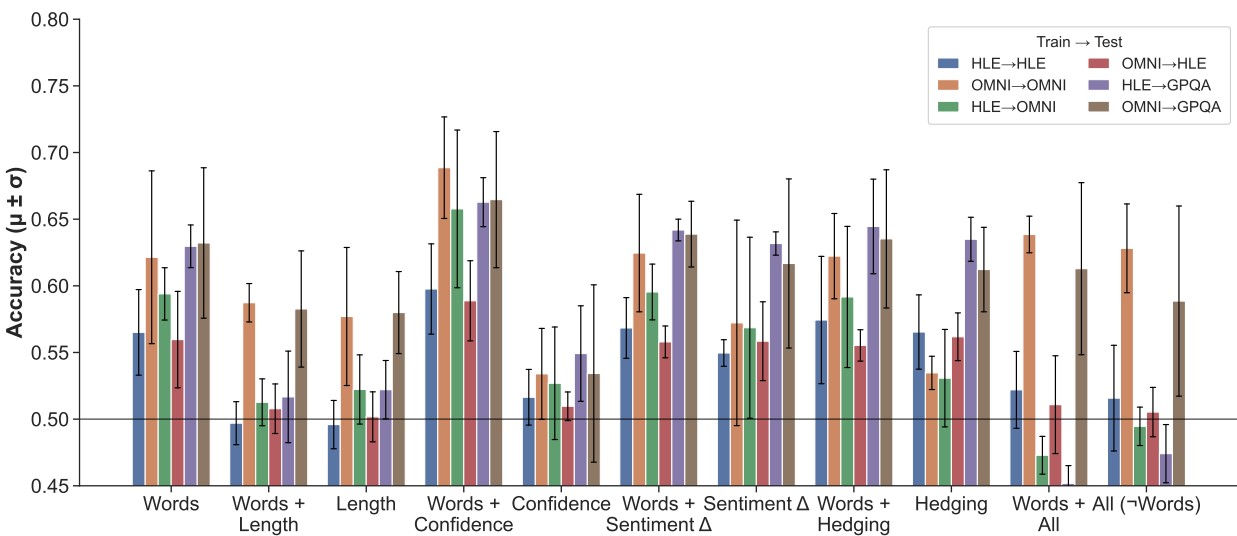

Figure 5: Each group of bars represents a different combination of input features: a bag-of-words model using 25 "harmful" words ("Words"), common surface features (reasoning length, sentiment, hedging), or their combinations. Neural networks were trained on imbalanced data but evaluated on balanced test sets to ensure that accuracy reflects genuine generalisation rather than label bias. Results are averaged across 30 random seeds, with error bars showing variability across runs. For each seed, the harmful-word lexicon was recompiled exclusively from the training data, preventing any information leakage. In-benchmark performance (blue: HLE → HLE, orange: Omni → Omni) remains consistently higher than cross-benchmark generalisation (green: HLE → Omni, red: Omni → HLE, purple: HLE → GPQA, brown: Omni → GPQA). The GPQA benchmark was used exclusively as a test set due to its smaller size, which prevents consistent identification of lexical cues (e.g., words occurring at least 300 times). Adding sentiment or hedging features provided limited benefit, while incorporating reasoning length generally reduced both in- and cross-benchmark accuracy. Overall, a compact, seed-specific lexicon of 25 harmful words proved to be the most robust and generalizable predictor of answer correctness across all three benchmarks.

## 4 Discussion and Conclusion

Our results reveal a clear information hierarchy within CoT traces. Lexical uncertainty hints dominate: tokens such as *complicated*, *probably*, and *likely* reduce accuracy odds by up to 25 % relative to baseline (Fig.3). Intra-CoT sentiment volatility is weaker but complementary; a modest upward sentiment shift ($\Delta = 0.1$) or upward slope in emotion coincide with improved accuracy on Omni-MATH and GPQA, yet sentiment is uninformative on HLE (Fig.2). On the moderate difficulty benchmarks Omni MATH and GPQA we find a negative correlation between CoT length and answer accuracy, whereas no such link appears on the more demanding HLE set. This pattern matches earlier work reporting accuracy declines with longer CoTs on Omni MATH (Ballon et al., 2025) and the absence of this effect on highly challenging benchmarks (Shojaee et al., 2025; Opus & Lawsen, 2025).

This asymmetry can be operationalised into a simple yet effective heuristic: filter out responses whose CoT contains strong uncertainty words, regardless of their length or self-reported confidence. Even a small lexicon of such "harmful" words performs on par with, or better than, more elaborate confidence-based approaches. Specifically, training a binary classifier using only the 25 most consistently harmful non-lemmatised words achieves Matthews correlation coefficients (MCCs) of 0.151 when trained and tested on HLE (and 0.158 cross-evaluated on Omni-MATH), 0.219 when trained and tested on Omni-MATH (0.113 cross-evaluated on HLE), and up to 0.278 and 0.282 when evaluated on GPQA. In contrast, simple confidence-thresholding heuristics yield substantially lower correlations, underscoring that certain lexical cues alone can serve as robust indicators of reasoning accuracy. Strikingly, a lightweight rule, i.e. mark an answer wrong if any of the top 5 harmful words appear in its CoT, otherwise classify as correct, achieves MCC = 0.144 on HLE,

0.322 on Omni-Math, and 0.390 on GPQA, see Appendix Table 7. This suggests that a handful of carefully selected lexical cues can serve as a highly practical post-hoc filter. A further advantage of this technique is that it treats the CoT as plain text: no weight access, no second inference pass is necessary.

The contrast between DeepSeek-R1 and Claude 3.7 Sonnet might illuminate the role of RLHF stylistics. Claude tends to generate longer, optimism-biased CoTs yet shares the same lexical uncertainty fingerprint as R1. This suggests that affective style might be an alignment artefact, whereas the full chain may not mirror the true multi-step search in latent space; the presence of hedging words does correlate with lower internal confidence, which is in line with (Chen et al., 2025b).

### Limitations and future directions

In this paper, two English benchmarks were studied, so our findings might not hold up in languages with richer morphology and different hedging forms. Furthermore, our hedging list is fixed and English-focused, meaning we miss any new or non-English uncertainty phrases. The sentiment model we used (OpenAI o3-mini) brings its own calibration quirks. Using multiple raters or a human-scored baseline could smooth out existing noise. Additionally, our HLE grading depends on LLM–human agreement, which implies that the mislabels could hide subtle effects. A related limitation concerns Qwen-235B-Think, whose responses were evaluated exclusively by OpenAI's o4-mini model. While o4-mini offers improved calibration and grading robustness relative to o3-mini, the absence of human cross-validation means that potential judgment errors by the evaluator cannot be fully ruled out. Finally, correlation is not causation: those uncertainty markers could simply be more common around tougher, hidden problems in general.

Key avenues for future work include: (i) multilingual replication with adaptive hedging lexicons; (ii) adversarial prompting to force models to suppress hedging and test the causal impact on accuracy; (iii) prompt-sensitivity and "thinking-budget" analyses that vary CoT styles or prompt framing to assess the robustness of lexical and reasoning-length effects; (iv) examining how lexical uncertainty markers relate to item-level question hardness; and (v) to understand RLHF's impact on uncertainty expression, future work could track changes in hedging frequency and harmful token usage between SFT and RLHF checkpoints. This includes testing whether RLHF amplifies or suppresses the lexical signals that predict incorrect reasoning. Lastly, it is known that LLMs are often fine-tuned to avoid producing certain "harmful" words; in practice this may block or distort a reasoning chain that could otherwise have led to a truthful answer.

### Broader Impact Statement

This work shows that simple lexical cues in an LLM's Chain-of-Thought can help predict likely errors, offering a lightweight and model-agnostic way to improve calibration. Such signals could support safer deployment of reasoning models in high-stakes domains. Risks include adversarial evasion of these cues and false positives. It should be noted that these lexical cues could trigger Goodhart's law: if they become widely used for filtering, model providers or users may fine-tune systems to suppress the flagged words without improving the underlying reasoning. Such stylistic suppression would undermine the predictive value of our signals while leaving true error rates unchanged. Mitigation of these risks may involve combining our method with other calibration techniques and extending tests to multilingual and domain-diverse settings. Broader testing across languages and domains is needed to ensure robustness, prevent misuse, and identify effects that may differ outside the English reasoning benchmarks studied.

### Data Availability

Data has been made available at: `https://figshare.com/s/547fe1f1e5fcb96e54bf`

**Author Contributions**

**Acknowledgments**

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

# A   Accuracy Breakdown

Table 2: Accuracy (with count) for each model by HLE category, Omni-MATH tier, Omni-MATH domain, and GPQA subdomain. For questions in Omni-MATH that span multiple domains, each domain instance is counted. Tiers are defined according to (Ballon et al., 2025). For both DeepSeek-R1 and Claude 3.7 Sonnet, the highest accuracy in each benchmark is shown in **bold**, and the second-highest is underlined. While all three models perform well on similar mathematical sub-disciplines on Omni-MATH, their areas of strength differ on HLE and GPQA.

| Dataset | Subset | Category/Tier/Domain | DeepSeek-R1 (n) | Claude 3.7 Sonnet (n) | Qwen2.5-72B (n) |
|---|---|---|---|---|---|
| HLE | Category | Biology/Medicine | 7% (68) | 6% (66) | 10% (71) |
| | | Chemistry | 2% (63) | 5% (63) | 8% (64) |
| | | Computer Science/AI | 6% (138) | 6% (145) | 9% (151) |
| | | Engineering | **15%** (34) | 6% (33) | **15%** (34) |
| | | Humanities/Social Science | 5% (95) | 8% (90) | 6% (99) |
| | | Math | 9% (793) | **10%** (804) | 13% (840) |
| | | Other | 3% (95) | 2% (98) | 4% (98) |
| | | Physics | 5% (141) | 6% (139) | 9% (148) |
| Omni-MATH | Tier | Tier 1 | **82%** (1445) | **80%** (1445) | **80%** (1445) |
| | | Tier 2 | 70% (1304) | 64% (1304) | 62% (1304) |
| | | Tier 3 | 66% (1227) | 63% (1227) | 46% (1227) |
| | | Tier 4 | 68% (452) | 67% (452) | 35% (452) |
| Omni-MATH | Domain | Algebra | **78%** (2139) | **76%** (2139) | 65% (2139) |
| | | Applied Mathematics | 71% (805) | 65% (805) | 67% (805) |
| | | Calculus | 73% (128) | 66% (128) | 66% (128) |
| | | Discrete Mathematics | 58% (888) | 55% (888) | 42% (888) |
| | | Geometry | 64% (1018) | 60% (1018) | 53% (1018) |
| | | Number Theory | 74% (916) | 74% (916) | 53% (916) |
| | | Precalculus | 76% (87) | 75% (87) | **69%** (87) |
| GPQA | Subdomain | Astrophysics | 77% (13) | 77% (13) | 100% (13) |
| | | Chemistry (general) | 65% (20) | 80% (20) | 80% (20) |
| | | Condensed Matter Physics | **100%** (1) | **100%** (1) | **100%** (1) |
| | | Electromagnetism and Photonics | 83% (6) | 67% (6) | 83% (6) |
| | | Genetics | 50% (4) | 50% (4) | 50% (4) |
| | | High-energy particle physics | 86% (14) | **100%** (14) | 93% (14) |
| | | Inorganic Chemistry | **100%** (1) | **100%** (1) | **100%** (1) |
| | | Molecular Biology | 60% (15) | 73% (15) | 73% (15) |
| | | Optics and Acoustics | **100%** (1) | **100%** (1) | **100%** (1) |
| | | Organic Chemistry | 35% (72) | 62% (72) | 61% (72) |
| | | Physics (general) | 79% (19) | 90% (19) | 95% (19) |
| | | Quantum Mechanics | 88% (25) | 88% (25) | 92% (25) |
| | | Relativistic Mechanics | 86% (7) | 86% (7) | 86% (7) |

## B    Calibration Error

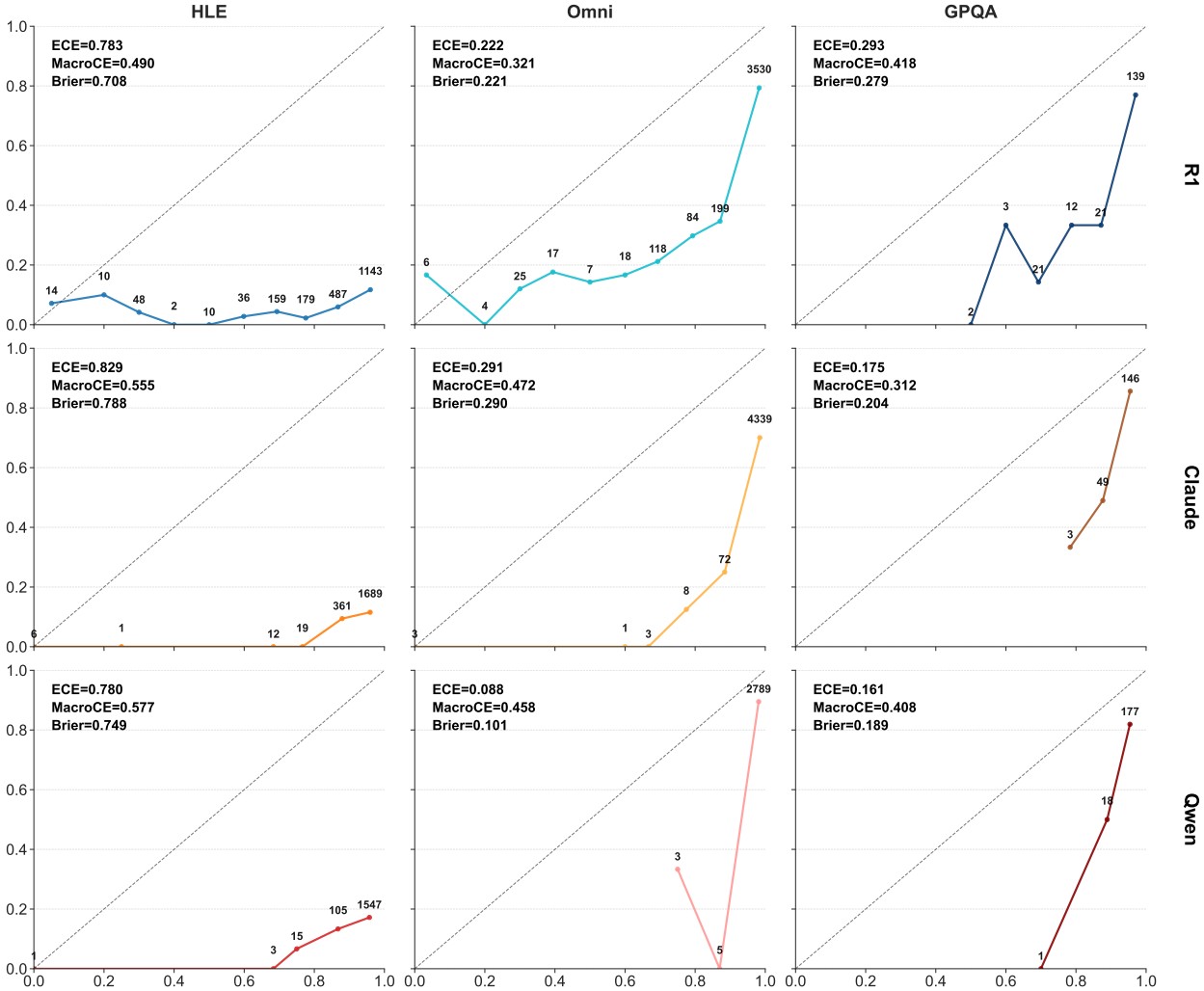

Figure 6: Calibration curves for DeepSeek-R1, Claude 3.7 Sonnet, and Qwen-235B-Think across the HLE, Omni-MATH, and GPQA benchmarks. Each panel compares self-reported confidence (x-axis) with actual accuracy (y-axis). Values above the diagonal indicate underconfidence, while those below indicate overconfidence. All models are consistently confident in their predictions as illustrated by the large amount of observations clustered in the right most dots on each figure.

## C   Booster Words

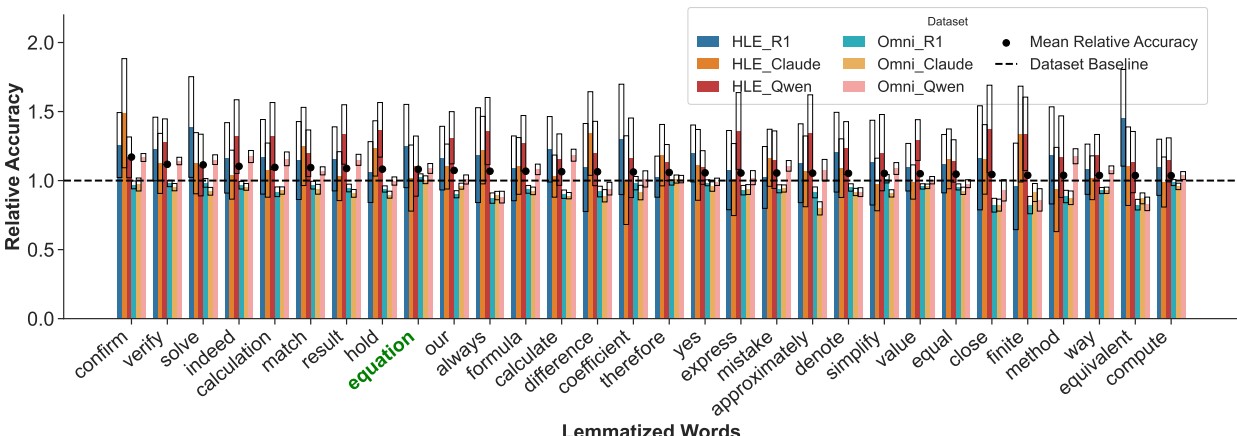

Figure 7: Relative accuracy of the 30 most beneficial lemmatised words across datasets and models. Words characteristic of mathematical reasoning (e.g., solve, therefore, equation) are generally associated with higher accuracy, but only "equation" (in green) appears universally beneficial across all benchmarks. This likely reflects the STEM orientation of the evaluated datasets and the models' relatively strong performance in mathematically focused domains. Error bars indicate 95% confidence intervals estimated via Monte Carlo resampling from the binomial distribution.

## D ROC curves

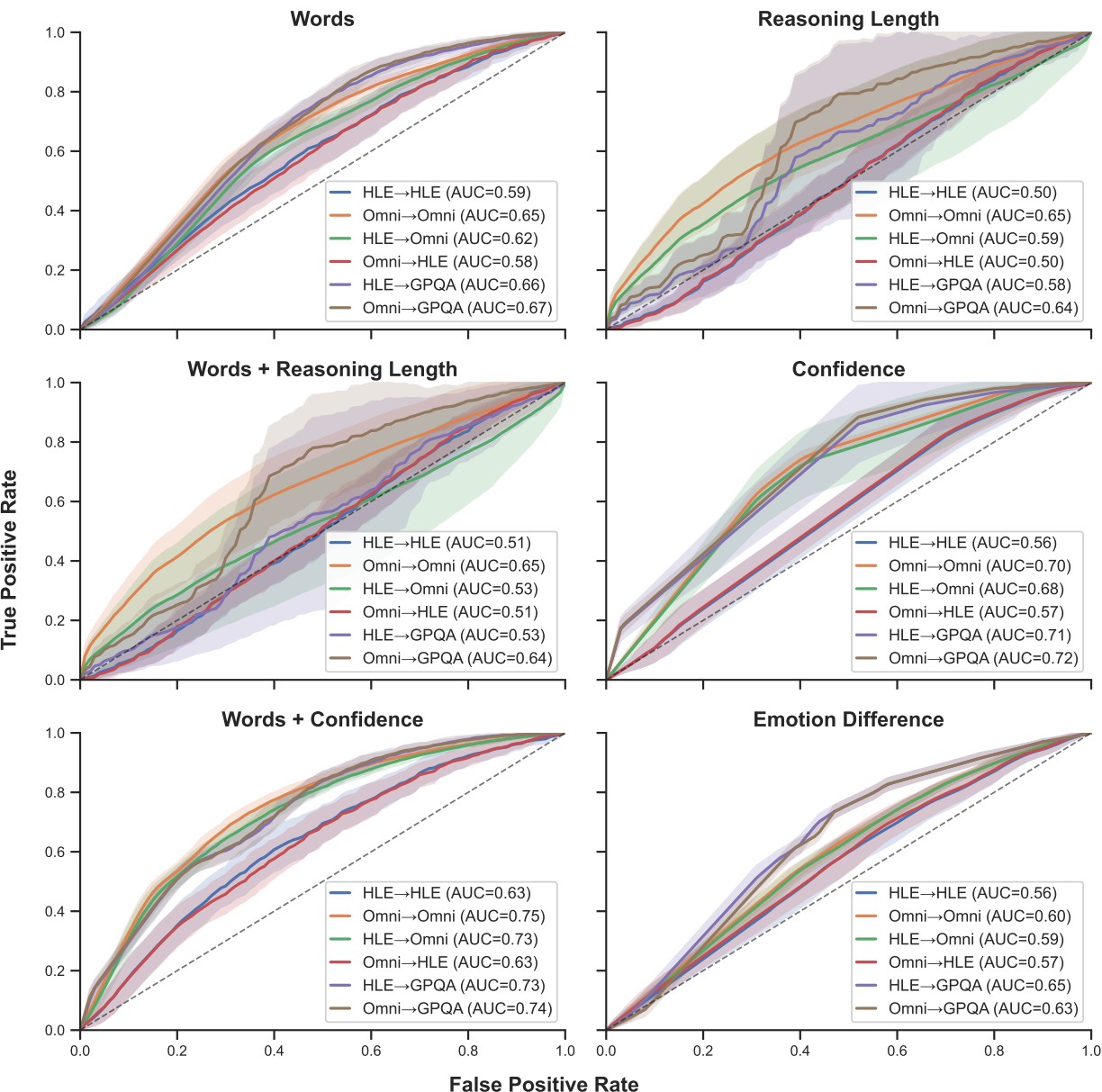

Figure 8

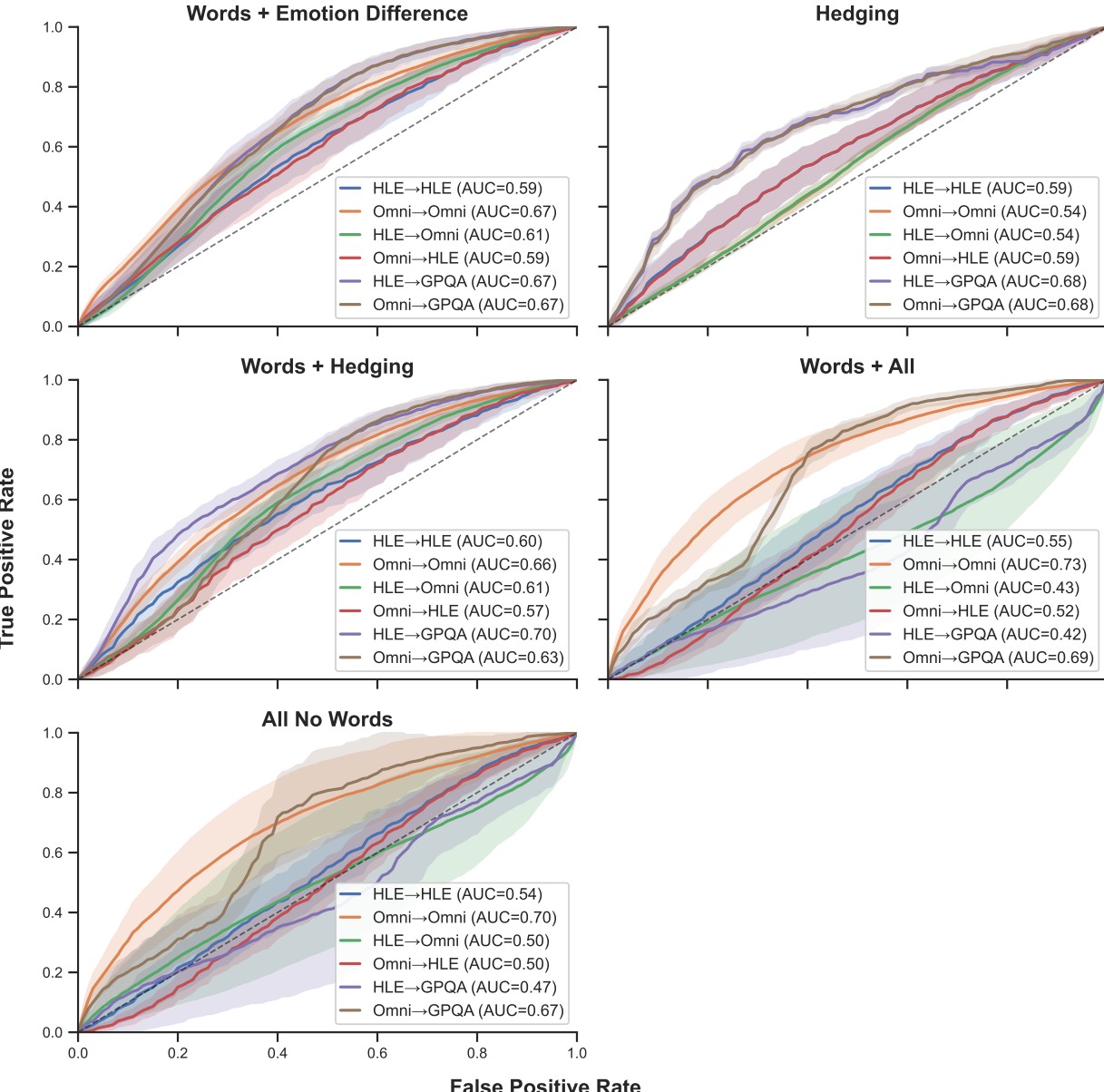

Figure 9: ROC curves across 11 experimental settings evaluating neural network classifiers for answer correctness. Each curve displays the mean true/false positive rates and 95% confidence interval (shaded region) across 30 random seeds. The models are trained and evaluated on balanced data for robust comparison. Area under the curve (AUC) values are reported in the legend. Models that use "harmful" words and abstain from using reasoning length perform the best and most consistently.

# E Consistently Harmful Words Used in Training

Table 3: General list of 25 non-lemmatised words most strongly and consistently associated with reduced accuracy ("harmful" words) for HLE and Omni-MATH. GPQA was not used to build this general lexicon due to its smaller corpus size, which prevents the identification of robust lexical effects.

| Harmful Words for Accuracy | | | | |
|---|---|---|---|---|
| positions | based | likely | probably | called |
| structure | position | depending | certain | direction |
| complex | often | four | per | else |
| related | group | complicated | too | making |
| similar | unless | system | correspond | looking |

# F    All Consistently Harmful Words

Table 4: List of all consistently "harmful" non-lemmatised words, including their 95% confidence interval offsets (in parentheses) and total word frequencies [in brackets]. The table includes all HLE, Omni-MATH, and Qwen model variants. GPQA was excluded due to insufficient token frequency (most words < 300 occurrences), which prevents stable accuracy estimation.

| Lemmatized Word | Mean Rel. Acc. | HLE-R1 | HLE-Claude | HLE-Qwen | Omni-R1 | Omni-Claude | Omni-Qwen |
|---|---|---|---|---|---|---|---|
| positions | 0.74 | 0.57 (0.28) [327] | 0.81 (0.27) [414] | 0.70 (0.23) [317] | 0.78 (0.04) [951] | 0.78 (0.04) [1142] | 0.81 (0.06) [738] |
| based | 0.80 | 0.76 (0.20) [807] | 0.93 (0.16) [1426] | 0.73 (0.21) [420] | 0.72 (0.04) [1099] | 0.83 (0.03) [2179] | 0.83 (0.07) [444] |
| likely | 0.80 | 0.66 (0.18) [849] | 0.92 (0.19) [981] | 0.83 (0.17) [778] | 0.73 (0.04) [1336] | 0.76 (0.04) [1117] | 0.91 (0.06) [664] |
| probably | 0.81 | 0.81 (0.25) [533] | 0.76 (0.22) [631] | 0.86 (0.15) [974] | 0.78 (0.05) [663] | 0.75 (0.05) [876] | 0.88 (0.05) [993] |
| called | 0.81 | 0.75 (0.30) [311] | 0.80 (0.25) [460] | 0.77 (0.22) [378] | 0.80 (0.06) [464] | 0.83 (0.07) [425] | 0.91 (0.09) [306] |
| structure | 0.82 | 0.78 (0.20) [785] | 0.92 (0.19) [903] | 0.91 (0.20) [562] | 0.78 (0.04) [968] | 0.77 (0.04) [1235] | 0.77 (0.07) [418] |
| position | 0.83 | 0.86 (0.28) [431] | 0.83 (0.24) [549] | 0.81 (0.20) [457] | 0.82 (0.04) [992] | 0.80 (0.04) [1265] | 0.82 (0.06) [843] |
| depending | 0.83 | 0.84 (0.28) [484] | 0.85 (0.24) [470] | 0.90 (0.27) [307] | 0.82 (0.04) [1029] | 0.79 (0.05) [720] | 0.78 (0.08) [420] |
| certain | 0.83 | 0.91 (0.19) [1067] | 0.90 (0.19) [978] | 0.84 (0.21) [441] | 0.77 (0.03) [1624] | 0.81 (0.04) [1392] | 0.75 (0.10) [317] |
| direction | 0.83 | 0.87 (0.31) [373] | 0.87 (0.24) [550] | 0.82 (0.22) [410] | 0.77 (0.05) [655] | 0.79 (0.05) [1020] | 0.88 (0.06) [720] |
| complex | 0.83 | 0.76 (0.23) [672] | 0.94 (0.18) [1135] | 0.90 (0.21) [455] | 0.74 (0.04) [1271] | 0.84 (0.03) [1986] | 0.82 (0.05) [1272] |
| often | 0.84 | 0.83 (0.26) [492] | 0.80 (0.24) [646] | 0.86 (0.20) [536] | 0.75 (0.06) [555] | 0.81 (0.07) [329] | 0.97 (0.07) [542] |
| four | 0.84 | 0.76 (0.25) [533] | 0.75 (0.29) [359] | 0.87 (0.23) [419] | 0.86 (0.04) [1285] | 0.87 (0.04) [993] | 0.92 (0.05) [1012] |
| per | 0.84 | 0.90 (0.25) [621] | 0.73 (0.26) [398] | 0.89 (0.19) [678] | 0.81 (0.05) [868] | 0.81 (0.06) [431] | 0.89 (0.06) [804] |
| else | 0.85 | 0.88 (0.29) [513] | 0.79 (0.26) [454] | 0.92 (0.19) [500] | 0.85 (0.04) [970] | 0.80 (0.05) [754] | 0.83 (0.07) [602] |
| related | 0.86 | 0.90 (0.21) [869] | 0.96 (0.19) [982] | 0.85 (0.16) [828] | 0.82 (0.04) [1389] | 0.80 (0.04) [1193] | 0.81 (0.06) [799] |
| group | 0.86 | 0.66 (0.26) [423] | 0.94 (0.29) [390] | 0.89 (0.24) [401] | 0.88 (0.06) [534] | 0.86 (0.06) [553] | 0.90 (0.08) [460] |
| complicated | 0.86 | 0.92 (0.20) [1002] | 0.84 (0.20) [668] | 0.84 (0.22) [443] | 0.87 (0.03) [2219] | 0.86 (0.03) [1881] | 0.81 (0.07) [500] |
| too | 0.87 | 0.75 (0.17) [970] | 0.95 (0.18) [1048] | 0.90 (0.19) [634] | 0.81 (0.03) [1955] | 0.85 (0.03) [2246] | 0.92 (0.04) [1255] |
| making | 0.87 | 0.69 (0.22) [607] | 0.98 (0.16) [1326] | 0.86 (0.26) [303] | 0.79 (0.04) [990] | 0.91 (0.03) [2972] | 0.97 (0.08) [450] |
| similar | 0.87 | 0.97 (0.21) [947] | 0.91 (0.21) [817] | 0.91 (0.18) [684] | 0.79 (0.03) [1963] | 0.82 (0.04) [1278] | 0.80 (0.05) [1037] |
| unless | 0.87 | 0.87 (0.22) [825] | 0.83 (0.28) [404] | 0.96 (0.21) [504] | 0.85 (0.03) [1618] | 0.87 (0.05) [783] | 0.83 (0.06) [693] |
| system | 0.87 | 0.85 (0.27) [478] | 0.81 (0.22) [602] | 0.93 (0.22) [448] | 0.85 (0.04) [922] | 0.86 (0.04) [1110] | 0.92 (0.06) [707] |
| correspond | 0.87 | 0.90 (0.27) [465] | 0.86 (0.29) [390] | 0.96 (0.26) [339] | 0.82 (0.05) [747] | 0.82 (0.05) [654] | 0.85 (0.07) [542] |
| looking | 0.87 | 0.74 (0.24) [610] | 0.95 (0.17) [1321] | 0.74 (0.21) [381] | 0.91 (0.04) [878] | 0.91 (0.03) [2868] | 0.97 (0.09) [333] |
| specifically | 0.87 | 0.93 (0.27) [501] | 0.89 (0.16) [1227] | 0.91 (0.25) [368] | 0.85 (0.05) [617] | 0.85 (0.03) [1998] | 0.80 (0.07) [359] |
| part | 0.87 | 0.76 (0.17) [1124] | 0.87 (0.19) [974] | 0.91 (0.15) [1072] | 0.89 (0.03) [1729] | 0.83 (0.04) [1361] | 0.98 (0.04) [1220] |
| angle | 0.88 | 0.90 (0.25) [606] | 0.90 (0.18) [983] | 0.84 (0.17) [783] | 0.81 (0.04) [1210] | 0.84 (0.03) [2155] | 0.97 (0.05) [1069] |
| specific | 0.88 | 0.88 (0.18) [1057] | 0.88 (0.14) [1592] | 0.94 (0.15) [963] | 0.81 (0.03) [1842] | 0.86 (0.03) [2762] | 0.88 (0.05) [1256] |
| starts | 0.88 | 0.86 (0.30) [363] | 0.94 (0.32) [300] | 0.87 (0.24) [384] | 0.88 (0.05) [826] | 0.81 (0.05) [646] | 0.91 (0.06) [686] |
| new | 0.88 | 0.90 (0.29) [425] | 0.98 (0.23) [674] | 0.92 (0.25) [397] | 0.83 (0.05) [950] | 0.83 (0.04) [1573] | 0.81 (0.05) [752] |
| actual | 0.88 | 0.94 (0.24) [621] | 0.89 (0.24) [599] | 0.97 (0.25) [400] | 0.80 (0.04) [893] | 0.71 (0.05) [753] | 0.98 (0.07) [529] |
| needs | 0.88 | 0.85 (0.28) [449] | 0.94 (0.23) [642] | 0.98 (0.23) [415] | 0.80 (0.05) [874] | 0.86 (0.04) [1498] | 0.86 (0.06) [646] |
| something | 0.89 | 0.95 (0.20) [1066] | 0.94 (0.17) [1279] | 0.84 (0.20) [980] | 0.88 (0.03) [2004] | 0.85 (0.03) [2281] | 0.87 (0.04) [1331] |
| within | 0.89 | 0.95 (0.30) [439] | 0.97 (0.24) [622] | 0.95 (0.22) [432] | 0.82 (0.04) [890] | 0.83 (0.04) [945] | 0.84 (0.06) [652] |
| assuming | 0.90 | 0.89 (0.19) [970] | 0.92 (0.19) [930] | 0.99 (0.17) [892] | 0.82 (0.04) [1344] | 0.85 (0.03) [1694] | 0.91 (0.05) [927] |
| parts | 0.90 | 0.96 (0.31) [435] | 0.96 (0.29) [418] | 0.83 (0.24) [330] | 0.86 (0.05) [759] | 0.81 (0.06) [585] | 0.97 (0.07) [526] |
| needed | 0.90 | 0.89 (0.28) [456] | 0.97 (0.29) [470] | 0.96 (0.24) [378] | 0.78 (0.05) [689] | 0.87 (0.05) [638] | 0.96 (0.07) [493] |
| count | 0.91 | 0.89 (0.28) [456] | 0.94 (0.24) [543] | 0.94 (0.20) [540] | 0.87 (0.04) [1163] | 0.84 (0.04) [1294] | 0.96 (0.04) [1246] |
| corresponds | 0.91 | 0.95 (0.28) [476] | 0.99 (0.24) [591] | 0.95 (0.24) [423] | 0.82 (0.04) [947] | 0.84 (0.04) [1076] | 0.89 (0.06) [717] |
| being | 0.91 | 0.93 (0.21) [998] | 0.94 (0.16) [1307] | 0.96 (0.17) [833] | 0.84 (0.03) [1750] | 0.86 (0.03) [2294] | 0.93 (0.04) [1368] |
| through | 0.91 | 0.97 (0.23) [744] | 0.99 (0.17) [1278] | 0.82 (0.18) [572] | 0.86 (0.03) [1454] | 0.88 (0.03) [2472] | 0.96 (0.05) [967] |
| could | 0.92 | 0.93 (0.17) [1298] | 0.98 (0.16) [1425] | 0.93 (0.16) [883] | 0.87 (0.03) [2325] | 0.88 (0.03) [2562] | 0.93 (0.04) [1676] |
| work | 0.92 | 0.93 (0.24) [699] | 0.97 (0.19) [1112] | 0.87 (0.23) [400] | 0.87 (0.03) [1749] | 0.94 (0.03) [2833] | 0.94 (0.05) [1000] |
| ones | 0.92 | 0.99 (0.31) [493] | 0.99 (0.27) [534] | 0.97 (0.23) [476] | 0.82 (0.04) [1095] | 0.84 (0.04) [1160] | 0.91 (0.05) [870] |
| however | 0.92 | 0.99 (0.15) [1694] | 0.99 (0.15) [1562] | 0.99 (0.14) [1309] | 0.88 (0.02) [2987] | 0.87 (0.03) [2252] | 0.80 (0.05) [1216] |
| go | 0.92 | 0.94 (0.21) [803] | 0.99 (0.15) [1517] | 0.97 (0.16) [883] | 0.82 (0.03) [1583] | 0.91 (0.02) [3266] | 0.92 (0.04) [1714] |
| question | 0.93 | 0.98 (0.17) [1368] | 0.99 (0.16) [1420] | 0.97 (0.14) [1293] | 0.90 (0.03) [1600] | 0.85 (0.03) [2140] | 0.89 (0.03) [2123] |
| integers | 0.93 | 0.83 (0.32) [307] | 0.93 (0.31) [362] | 0.98 (0.27) [326] | 0.94 (0.03) [1587] | 0.97 (0.03) [1799] | 0.94 (0.04) [1619] |
| my | 0.93 | 0.99 (0.22) [763] | 1.00 (0.14) [1836] | 0.98 (0.20) [647] | 0.85 (0.04) [1394] | 0.94 (0.02) [3741] | 0.85 (0.05) [930] |
| starting | 0.94 | 0.99 (0.22) [680] | 0.97 (0.21) [893] | 0.96 (0.22) [481] | 0.94 (0.03) [1881] | 0.84 (0.03) [1856] | 0.93 (0.04) [1214] |
| side | 0.94 | 0.96 (0.28) [470] | 0.94 (0.28) [519] | 0.86 (0.20) [525] | 0.95 (0.03) [1731] | 0.94 (0.03) [1739] | 0.99 (0.04) [1760] |
| they | 0.94 | 0.97 (0.16) [1487] | 0.98 (0.15) [1482] | 0.96 (0.13) [1515] | 0.90 (0.02) [2873] | 0.90 (0.02) [3000] | 0.95 (0.03) [2643] |
| like | 0.95 | 0.97 (0.16) [1535] | 0.95 (0.15) [1658] | 0.99 (0.14) [1255] | 0.91 (0.02) [3036] | 0.90 (0.02) [3158] | 0.95 (0.04) [1771] |

Table 5: Average accuracies (mean $\pm$ standard deviation over 30 seeds) for domain-sensitivity experiments, comparing models trained on HLE and Omni-MATH datasets and evaluated on math and non-math subsets of the HLE test set. All evaluations were performed on balanced test sets.

| Train→Test | Words | Words + Reasoning Length | Confidence | Words + Confidence |
|---|---|---|---|---|
| HLE→HLE$_{math}$ | $0.526 \pm 0.056$ | $0.494 \pm 0.018$ | $0.515 \pm 0.039$ | $0.594 \pm 0.077$ |
| HLE→HLE$_{non\text{-}math}$ | $0.566 \pm 0.022$ | $0.492 \pm 0.019$ | $0.516 \pm 0.024$ | $0.580 \pm 0.032$ |
| Omni→HLE$_{math}$ | $0.513 \pm 0.073$ | $0.493 \pm 0.051$ | $0.511 \pm 0.041$ | $0.586 \pm 0.072$ |
| Omni→HLE$_{non\text{-}math}$ | $0.566 \pm 0.037$ | $0.507 \pm 0.025$ | $0.512 \pm 0.026$ | $0.578 \pm 0.034$ |

Table 6: Confusion matrices and Matthews Correlation Coefficients (MCC) for our best-performing individual feature ("harmful" words), trained using only the 25 most consistently harmful words as input features. Evaluations were conducted on balanced test sets, with results shown for a single random seed (seed = 2).

| Train→Test | Actual | Pred. 0 | Pred. 1 | MCC |
|---|---|---|---|---|
| HLE→HLE | 0 | 67 | 59 | 0.151 |
| | 1 | 48 | 78 | |
| HLE→Omni-MATH | 0 | 355 | 511 | 0.158 |
| | 1 | 226 | 640 | |
| HLE→GPQA | 0 | 78 | 90 | 0.278 |
| | 1 | 34 | 134 | |
| Omni-MATH→Omni-MATH | 0 | 368 | 498 | 0.219 |
| | 1 | 191 | 675 | |
| Omni-MATH→HLE | 0 | 59 | 67 | 0.113 |
| | 1 | 45 | 81 | |
| Omni-MATH→GPQA | 0 | 80 | 88 | 0.282 |
| | 1 | 35 | 133 | |

Table 7: Confusion matrices and Matthews Correlation Coefficients (MCC) for two heuristic methods evaluated on the HLE, Omni-Math, and GPQA datasets using a balanced test set. The 5 Harmful Words heuristic predicts failure if any of the five most consistently harmful words appear in the reasoning text. The Confidence heuristic uses self-reported confidence to probabilistically predict correctness by flipping coin, weighted by its reported confidence level, to assign a grade of either 0 or 1. Our results illustrate that the word-based heuristic consistently outperforms the probabilistic confidence-based method.

| Heuristic | Dataset | Actual | Pred. 0 | Pred. 1 | MCC |
|---|---|---|---|---|---|
| 5 Most Harmful Words | HLE | 0 | $59.9 \pm 4.7$ | $66.1 \pm 4.7$ | $0.144 \pm 0.058$ |
| | | 1 | $42.1 \pm 4.7$ | $83.9 \pm 4.7$ | |
| | Omni-Math | 0 | $302.5 \pm 9.1$ | $246.5 \pm 9.1$ | $0.322 \pm 0.025$ |
| | | 1 | $129.8 \pm 10.8$ | $419.2 \pm 10.8$ | |
| | GPQA | 0 | $17.6 \pm 2.4$ | $15.4 \pm 2.4$ | $0.390 \pm 0.100$ |
| | | 1 | $5.4 \pm 1.9$ | $27.6 \pm 1.9$ | |
| Confidence | HLE | 0 | $10.5 \pm 3.6$ | $115.5 \pm 3.6$ | $0.051 \pm 0.067$ |
| | | 1 | $7.1 \pm 2.5$ | $118.9 \pm 2.5$ | |
| | Omni-Math | 0 | $35.2 \pm 4.4$ | $513.8 \pm 4.4$ | $0.115 \pm 0.020$ |
| | | 1 | $10.2 \pm 2.9$ | $538.8 \pm 2.9$ | |
| | GPQA | 0 | $3.6 \pm 1.7$ | $29.4 \pm 1.7$ | $0.119 \pm 0.105$ |
| | | 1 | $1.5 \pm 1.1$ | $31.5 \pm 1.1$ | |

