# OpenReview forum: "Lexical Hints of Accuracy in LLM Reasoning Chains"
_TMLR — Rejected by TMLR_

### Review · Reviewer_bq6N · 2025-09-15

**Summary Of Contributions:**

This paper studies whether certain features of the model-generated explicit Chain-of-Thought can provide signals to measure model confidence, including the length of COT, the intra-CoT sentiment volatility, and some lexicographic hints. The authors conduct experiments on both open-source and closed-source models on HLE and Omni-MATH. Experiments show that these features about CoT can serve as a lightweight signal to measure the model confidence given certain conditions, e.g., CoT length only works for intermediate-difficulty benchmarks.

Strength:

1.	Studying effective measures of the model's internal confidence is important for both understanding the mechanism of CoT and deploying models in the downstream applications.

2.	The difference of the signals in different benchmarks, e.g., the feature of length on Omni-MATH and HLE, is an interesting finding and worth further investigation.

3.	The good performance of lexical cues can provide insights for future work studying the model CoT from the psychological perspective.

Weakness:

1.	The experiment setting is a bit arbitrary and not well-connected with existing research. (1) For baseline confidence elicitation methods: only directly reported confidence is considered (to the best of the reviewer’s knowledge, [1] should be, but not, cited for the self-reported confidence). Various other methods not important ways to measure the model confidence they are not considered, including but not limited to verbalized confidence with better prompts [2] and P(True) [3]. Since open-source models are also considered, the reviewer would also suggest including the token probabilities of the answer as a widely considered baseline. (2) For task choices, the current choices of HLE and Omni-MATH are a bit arbitrary since they are different in both type and hardness. It would be good if the authors could further control the study with general/math tasks at different hardness levels (e.g., 2x3 tasks) to further remove the confounders on what works and what does not. (3) For metrics, in Section 2.1 of this paper, the authors introduce the measure of calibration error. However, there is no clear definition of the specific metric. In the previous work, such as [2], ECE (expected calibration error) is commonly considered as the metric for calibration error [4]. Other variants, such as MacroCE [5] and Brier Score [6] can also be considered to show different characteristics of the confidence calibration distribution, as done in [7]. (4) For base models, the reviewer acknowledges the usage of one open-source and one closed-source model. It would be good if the authors could provide further model ablation on a subset of the task examples, e.g., comparing the performance of 8B, 7B models, and best commercial models (e.g., Gemini 2.5 pro), besides the current ones.

2.	The main method that was proposed in the paper is a linguistic clue. However, there is already a line of work revealing the relevance between model behavior and uncertainty expressions [8][9], which limits the novelty and contribution of the current work. The reviewer would suggest the authors to revise the schema of the uncertainty expression and improve upon the existing closely relevant work.

[1] https://arxiv.org/abs/2205.14334

[2] https://arxiv.org/pdf/2305.14975

[3] https://arxiv.org/abs/2207.05221

[4] https://proceedings.mlr.press/v70/guo17a.html

[5] https://arxiv.org/abs/2205.12507

[6] https://www.google.com/books/edition/Monthly_Weather_Review/jnbpAAAAMAAJ?hl=en&gbpv=1&dq=Glenn+W.+Brier.+1950.+Verification+of+forecasts+expressed+in+terms+of+probability.+Monthly+Weather+Review.+&pg=RA1-PA1&printsec=frontcover

[7] https://arxiv.org/pdf/2402.17124

[8] https://arxiv.org/abs/2302.13439

[9] https://arxiv.org/abs/2401.06730

**Audience:**

Yes

**Audience Explanation:**

Yes, this paper would be interesting to people studying CoT, reasoning tasks, and confidence calibration

**Broader Impact Concerns:**

There is no concern about the ethical implications

**Claims And Evidence:**

No

**Claims Explanation:**

No, the evidence is accurate, but the experiment settings (tasks, models, and metrics) can be further improved to make the conclusion clearer.

**Requested Changes:**

The suggested changes are as follows:

1.	Set up the experiments with more relevance to the rich existing work on model confidence calibration, in terms of the confidence extraction methods, metrics, and tasks, etc, as suggested in the weakness part.

2.	Build the story upon existing work, studying the relevance between model behavior and uncertainty expressions, as listed in the weakness part

3.	An interesting direction to add to the novelty of the work is to link to uncertainty/confidence calibration to work that is relevant to point-wise hardness, e.g., [8]. If the authors can show the different effectiveness of the linguistic clues for individual questions (instead of tasks) with different hardness, the conclusion and findings can be more exciting.

---

> ### Author Response · Authors · 2025-10-10
> **Revisions made to Lexical Hints of Accuracy in LLM Reasoning Chains**
>
> Below we add a case by case answer to Referees comments, in which we also highlight the changes we have made to the manuscript.
>
> The experiment setting is a bit arbitrary and not well-connected with existing research. See Remarks (1)-(4)
>
>
> -> (1) We now explicitly cite the relevant prior work on verbalized confidence elicitation and clarify that our prompting format closely follows the HLE benchmark. We also disucss comlimentary token-probability–based confidence measures (e.g., P(True)) in the manuscript. Unfortunately these require logit-level access, which is unavailable for closed-source models such as Claude 3.7 Sonnet.
> -> (2) To reduce potential confounding by task domain or difficulty, we have expanded our evaluation from a 2×2 to a 3×3 benchmark–model design, introducing a new non-mathematical reasoning benchmark (GPQA) and an additional model family (Qwen). Moreover, we conducted an auxiliary domain-sensitivity analysis within HLE, comparing model performance on mathematical versus non-mathematical questions while jointly controlling for CoT length (Table 5). We find that the lexical-only models performed worse on math than on non-math items (0.52 vs. 0.56 accuracy). Adding reasoning length levels this performance but at the cost of accuracy. While adding confidence also levels out performance in both domains but improves accuracy. p. 10
> -> (3) In Section 2.1 and Appendix Figure 6, we now provide a clear definition of the calibration metric used—Expected Calibration Error (ECE) and report complementary calibration diagnostics (MacroCE and Brier Score).
> -> (4) While broader model-scale ablations (e.g., 7B/8B variants or Gemini 2.5 Pro) would further strengthen generalization, they were beyond the present study’s computational and time constraints. Nevertheless, the expanded 3×3 configuration confirms that our main results—lexical features being the most robust predictors, with CoT length providing limited incremental value—remain stable across benchmarks and model families.
>
> The main method that was proposed in the paper is a linguistic clue. However, there is already a line of work revealing the relevance between model behavior and uncertainty expressions [8][9], which limits the novelty and contribution of the current work. The reviewer would suggest the authors to revise the schema of the uncertainty expression and improve upon the existing closely relevant work.
>
> -> We have included this relevant line of research in our manuscript. Prior studies such as [8]and [9] indeed demonstrate that linguistic uncertainty markers—e.g., epistemic weakeners and strengtheners—can influence or reflect model confidence. However, these works focus primarily on prompt-level or output-level manipulations of non-reasoning models, analysing or injecting uncertainty expressions directly into model responses to test their causal effects on calibration.
> In contrast, our study examines naturally emerging lexical uncertainty markers within models’ chain-of-thought (CoT) reasoning, across multiple benchmarks and model families. Rather than modifying prompts or outputs, we quantify how spontaneous hedging and related linguistic cues correlate with final correctness, providing a large-scale observational complement to prior intervention-based studies. This distinction—focusing on reasoning traces rather than surface responses—is now explicitly clarified in the revised manuscript (in the introduction and on p.6).
>
> An interesting direction to add to the novelty of the work is to link to uncertainty/confidence calibration to work that is relevant to point-wise hardness, e.g., [8]. If the authors can show the different effectiveness of the linguistic clues for individual questions (instead of tasks) with different hardness, the conclusion and findings can be more exciting.
>
> -> This is indeed an interesting extension, as it would allow testing whether linguistic cues (e.g., hedging or boosters) vary systematically with instance-level difficulty. However, measuring individual question hardness in a benchmark-independent and model-agnostic manner is nontrivial—it typically requires per-item accuracy distributions or difficulty calibration across models, which is beyond the present study’s scope. We view this as a promising direction for future research and have added a note to this effect in the manuscript.

---

### Review · Reviewer_ExZd · 2025-09-20

**Summary Of Contributions:**

The paper compares different markers from LLM chain-of-thought (CoT) traces and reports correlations between these markers and the correctness of the final answers. The experiments are performed with Claude 3.7 Sonnet and DeepSeek-R1, on the Omni-MATH and the Humanity's last exam (HLE) benchmark. As for markers, the paper reports results on CoT length, sentiment volatility, and lexical hints of uncertainty.

The results show that all markers are correlated with answer correctness on Omni-MATH, but on HLE only the lexicographic analysis is correlated with answer correctness. Finally, a simple rule based on the presence of any of a set of five words is shown to perform well on the presented models and benchmarks.

**Additional Comments:**

1. The calibration error reported in Table 1 seems pretty bad. It might be interesting to provide more details on this (e.g. histograms for correct vs. incorrect answers, maybe with a breakdown by category).

2. The lexicographic analysis discards words that appeared less than 300 times. It would be interesting to report how often the chosen words did appear in the CoTs.

**Audience:**

Yes

**Audience Explanation:**

I think there is a substantial part of the audience of TLMR that is interested in understanding how CoT markers are correlated to answer correctness, and the paper shows interesting findings how some previously reported markers (like CoT length) might not work with all benchmarks, and a simple that lexicographic analysis might work even in those cases where CoT length fails.

**Claims And Evidence:**

Yes

**Claims Explanation:**

The results from the paper demonstrate that there is a solid correlation for all markers with Omni-MATH, but for the much more challenging HLE CoT length and sentiment analysis are only weakly correlated. These results are confirmed in a second experiment, where a binary classifier is trained on various feature combinations and evaluated with respect to its usefulness for predicting answer correctness.

The findings are though somewhat limited in scope, because they are only verified on two LLMs, and moreover, most of the findings only apply to a single test (HLE). So while the results do lend support for the claims, the more surprising findings (such as the missing correlation of CoT length and answer correctness) are based on findings from a single benchmark, and they might not generalize for LLM CoTs on difficult benchmarks in general (as suggested by the authors).

**Requested Changes:**

1. I would like to see better evidence for the claim that "CoT length predicts correctness only in the intermediate-difficulty benchmarks, i.e., inside the model's demonstrated capability, but still below saturation". From the results in the paper I only see that CoT length is a reliable marker for answer correctness in Omni-MATH, but not for HLE. Is this a particularity of the HLE benchmark? Might CoT length only work with math-like benchmarks? In order to answer these questions it would be helpful to confirm the finding with other "medium" and "hard" benchmarks, or it might also be insightful to better understand how CoT length performs on parts of the presented benchmarks (e.g., maybe CoT length is indicative of answer correctness for the math question in HLE).

2. In Figure 3, I find the "empty bars" hard to read. I assume they indicate some kind of uncertainty on the bar heights. The authors should specify in the caption whether the horizontal lines on the bars correspond to standard deviation and standard error. Using error bars might make this also clearer visually. If the markers correspond to standard error, then the exact methodology of computation should be reported as well. Similarly, in other figures that have lineplots, it should be described what the shaded areas correspond to exactly.

---

> ### Author Response · Authors · 2025-10-10
> **Revisions made to Lexical Hints of Accuracy in LLM Reasoning Chains**
>
> Below we add a case by case answer to Referees comments, in which we also highlight the changes we have made to the manuscript.
>
> I would like to see better evidence for the claim that "CoT length predicts correctness only in the intermediate-difficulty benchmarks, i.e., inside the model's demonstrated capability, but still below saturation". From the results in the paper I only see that CoT length is a reliable marker for answer correctness in Omni-MATH, but not for HLE. Is this a particularity of the HLE benchmark? Might CoT length only work with math-like benchmarks? In order to answer these questions it would be helpful to confirm the finding with other "medium" and "hard" benchmarks, or it might also be insightful to better understand how CoT length performs on parts of the presented benchmarks (e.g., maybe CoT length is indicative of answer correctness for the math question in HLE).
>
> -> We have clarified this point and extended our analysis to include the GPQA-Diamond benchmark, which has a similar saturation level as Omni-MATH but is not math-specific. As shown in Figure 1 and Table 5, CoT length is indeed predictive of correctness for Omni-MATH and GPQA but not for HLE, confirming that the effect generalises beyond mathematical domains. The added domain-sensitivity analysis (Table 5) further demonstrates that this pattern is not driven by math-related content within HLE: when controlling for problem type and CoT length, lexical predictiveness persists while CoT length alone loses discriminative power.
>
> In Figure 3, I find the "empty bars" hard to read. I assume they indicate some kind of uncertainty on the bar heights. The authors should specify in the caption whether the horizontal lines on the bars correspond to standard deviation and standard error. Using error bars might make this also clearer visually. If the markers correspond to standard error, then the exact methodology of computation should be reported as well. Similarly, in other figures that have lineplots, it should be described what the shaded areas correspond to exactly.
>
> -> We have clarified the meaning of the black boxes in the caption of the figure as follows: [...] Words that consistently reduce accuracy are highlighted in red, while dataset means are indicated by dashed lines, mean relative accuracies by black dots, and 95\% confidence intervals (estimated via Monte Carlo resampling from the binomial distribution) by black rectangles.[...]. Furthermore, we have also improved the readability of the figure.
>
> Calibration error in Table 1 seems high; provide more detail (e.g., histograms for correct vs.\ incorrect, breakdown by category).
>
> -> We have added a full breakdown of the calibration per model and benchmark in the appendix. Which illustrates that most models judge their output as highly confident regardless of correctness. The full overview of which can now be found in Appendix \textbf{B} on p. 17
>
> Lexicographic analysis discards words appearing $<300$ times; report frequencies for chosen words.
>
> -> We have added the word frequencies to the breakdown table on p. 22.

---

### Review · Reviewer_vaYJ · 2025-09-28

**Summary Of Contributions:**

## Summary
The paper tests whether simple CoT surface features (length, sentiment volatility, and lexical hedges) are predictive of answer uncertainty and correctness. Experiments are conducted using Claude 3.7 Sonnet and DeepSeek-R1 (as reasoning traces are accessible) on Omni-MATH and HLE.

The authors provide experiments that show the following:
- Length and sentiment volatility provide weak or benchmark-specific signals, while
- Lexical cues (e.g., words "guess", "stuck") are general indicators of when a model is likely wrong.

The manuscript frames these features as a lightweight, post-hoc uncertainty signal that can help flag low-trust answers, especially when the model's own self-reported confidence is unreliable.

## Strengths

- Focuses on easy-to-extract signals that don’t require logprobs, extra sampling, or weight access. All of this would make mitigation approaches (i.e., predictor/guardrail models) practical for deployment for models that provide access to CoT
- Nice discussion of broader impacts and the risk of adversarial evasion by style-shifting

## Weaknesses
- *Potential leakage / feature selection ambiguity in experimental design* Section 3.4 trains predictors using the 25 most consistently “harmful” non-lemmatized words. The paper does not clearly state whether discovery of these 25 words was performed only on training folds (nested within each split) or on the full dataset prior to model training/evaluation. If the latter, the pipeline leaks test labels into feature selection and can inflate in- and cross-benchmark performance. This needs to be clarified and, if not run in a clean way, re-run with nested CV to select informative words from the lexicons.

- *Limited external validity* Conclusions are drawn from 2 English benchmarks and 2 models.  Without a larger sample of datasets, it's difficult to draw conclusions on how robust these findings are. There is also room for considerable variability due to specific prompts used here. The authors note some of these limitations in the Limitations section, but the core framing the main contribution (“lexical uncertainty markers generalize best”) is too strong without broader replication.

- It’s not clear how much lexical predictiveness survives after controlling for problem type/domain (e.g., math vs. non-math within HLE) and CoT length simultaneously.
  - Booster words (fig 6) seem largely skewed towards mathematical tokens, which makes sense given the limited number of datasets used.

**Audience:**

Yes

**Audience Explanation:**

There is clear interest in post-hoc calibration and CoT-based monitoring and lightweight, CoT-based signals are attractive in practice.

**Broader Impact Concerns:**

It would be nice to comment on the following and how future methods development should consider these issues:

- **Evasion/gameability**: If systems down-weight answers containing certain words, models may be prompted or fine-tuned to suppress those words without improving reasoning quality.

- **Over-filtering**: A few “harmful” words might unfairly block correct answers in some domains e.g., scientific writing with appropriate hedging.

Some risks are acknowledged, but a brief mitigation plan (ensemble of signals, multilingual checks, adversarial stress tests) should be added to the Broader Impact section.

**Claims And Evidence:**

No

**Claims Explanation:**

The qualitative trend (that certain lexical cues correlate with errors and that length is not universally predictive) is supported on the two chosen datasets. But the methodological ambiguity around how the top 25 words were selected (e.g., figure 5 results confounded by dataset leakage) and the narrow scope (2 datasets, 2 models, 1 language) mean the claims about generalizable lexical proxies are not convincingly established yet.

**Requested Changes:**

## Critical changes for acceptance

- Eliminate leakage risk. If harmful word selection was not conducted in a clean way, re-run the harmful word (feature) discovery with nested cross-validation or perform selection on training folds only and report the drop, if any, in performance.

- Add broader validation. Include at least two additional benchmarks (e.g., GPQA (Graduate-Level Google-Proof Q&A), some non-math reasoning set) and one additional model family. A small non-English slice (even if limited) would materially strengthen the generalization claim.

## Other Improvements (not critical)
- Providing prompt sensitivity studies (different CoT styles or thinking budgets) to show robustness of these findings would make the papar stronger

---

> ### Author Response · Authors · 2025-10-10
> **Revisions made to Lexical Hints of Accuracy in LLM Reasoning Chains**
>
> Below we add a case by case answer to Referees comments, in which we also highlight the changes we have made to the manuscript.
>
> Potential leakage / feature selection ambiguity in experimental design Section 3.4 trains predictors using the 25 most consistently “harmful” non-lemmatized words. The paper does not clearly state whether discovery of these 25 words was performed only on training folds (nested within each split) or on the full dataset prior to model training/evaluation. If the latter, the pipeline leaks test labels into feature selection and can inflate in- and cross-benchmark performance. This needs to be clarified and, if not run in a clean way, re-run with nested CV to select informative words from the lexicons.
>
> -> Labels were indeed leaking into the test set. This has now been rerun in a clean and leakfree manner, the details of which have been specified on p.9-10.
>
> Conclusions are drawn from 2 English benchmarks and 2 models. Without a larger sample of datasets, it's difficult to draw conclusions on how robust these findings are. There is also room for considerable variability due to specific prompts used here. The authors note some of these limitations in the Limitations section, but the core framing the main contribution (“lexical uncertainty markers generalize best”) is too strong without broader replication.
>
> -> We agree that broader replication across languages, benchmarks, and model families would further strengthen the generality of our findings. However, many reasoning benchmarks are already near-saturated in performance, and only a limited number of reasoning-enabled models with accessible CoT traces are publicly available. Within these constraints, we expanded our evaluation from a 2×2 to a 3×3 benchmark–model matrix by adding a new benchmark (GPQA) and a new model family (Qwen).
>
> It’s not clear how much lexical predictiveness survives after controlling for problem type/domain (e.g., math vs. non-math within HLE) and CoT length simultaneously.
> Booster words (fig 6) seem largely skewed towards mathematical tokens, which makes sense given the limited number of datasets used.
>
> -> We have added a new domain-sensitivity analysis (Table 5) that jointly controls for problem type (math vs. non-math) and CoT length. The results show that lexical-only models actually perform worse on math than on non-math questions in HLE. When we included reasoning length in the feature vector length, this difference disappears, but overall accuracy drops to random guessing. We have added this nuance on p.10.
>
> Add broader validation. Include at least two additional benchmarks (e.g., GPQA (Graduate-Level Google-Proof Q\&A), some non-math reasoning set) and one additional model family. A small non-English slice (even if limited) would materially strengthen the generalization claim.
>
> -> We expanded our evaluation from a 2×2 benchmark–model design (HLE and Omni-MATH × Claude and DeepSeek) to a 3×3 setting by adding the GPQA-Diamond benchmark and the Qwen-235B-Think model family. GPQA provides a non-mathematical, graduate-level reasoning set situated between Omni-MATH and HLE in difficulty, thereby broadening both the domain and difficulty coverage. We also included an additional domain-sensitivity analysis within HLE (math vs. non-math) to further test the robustness of lexical and CoT-length predictors.
>
> Other improvement: prompt sensitivity studies (different CoT styles / thinking budgets) to show robustness.
>
> -> We agree that prompt-sensitivity and “thinking-budget” analyses (e.g., varying CoT styles or prompt framing) would provide valuable insights into the robustness of lexical and reasoning-length effects. However, such experiments are beyond the present study's scope. We envision that this question become even more relevant once more reasoning models are publicly available. We have added this suggestion to the limitations and future directions section of the manuscript.
>
> Some risks are acknowledged, but a brief mitigation plan (ensemble of signals, multilingual checks, adversarial stress tests) should be added to the Broader Impact section.
> Evasion/gameability: If systems down-weight answers containing certain words, models may be prompted or fine-tuned to suppress those words without improving reasoning quality.
> Over-filtering: A few “harmful” words might unfairly block correct answers in some domains e.g., scientific writing with appropriate hedging.
>
> ->We have expanded the Broader Impact section to explicitly discuss potential evasion and over-filtering effects. The revised text now notes that lexical cues could become susceptible to Goodhart’s law if models are fine-tuned to suppress them without improving reasoning quality, and that this may lead to unfair filtering of valid responses. We also added a brief mitigation plan outlining the importance of combining lexical cues with complementary calibration methods and extending robustness checks to multilingual and domain-diverse settings.

---

### Author Response · Authors · 2025-10-10
**Answer to all Referees**

We thank the reviewers for their careful reading of our manuscript and for their constructive and insightful comments, which have helped us to improve the paper substantially.

---

### Decision · Action_Editor_EpbY · 2025-11-20

**Recommendation:** Reject

**Audience:**

Yes

**Audience Explanation:**

please see above. in short: CoT is a dominant strategy in all existing LLMs but the manuscript, even after the efforts during the rebuttal, lacks enough rigor in the analysis to be ready for publication. I acknowledge the complexity of the problem, however, that does not make it alright to publish incomplete analysis.

**Claims And Evidence:**

No

**Claims Explanation:**

The paper shows that a lexicographic analysis of chain-of-thought (CoT) trails of thinking models provides markers that correlate with question answering accuracy. The authors further show that the length of CoT trails not always serves as a reliable marker. The proposed method is simple and seems to work relatively well in the examined setting.

But the paper only examines 3 thinking models and 3 benchmarks, and for example the claim that "CoT length predicts correctness only in the intermediate-difficulty benchmarks" (quote from the abstract of the paper) seems a bit thin, given that CoT length predicts correctness for two of three examined benchmarks (Omni-MATH and GPQA-diamond – but not HLE), and given that this result is somewhat surprising comparing to previous literature. This point was criticized by all reviewers. Note that the authors extended the results somewhat from the original two benchmarks by adding the results for GPQA-diamond.

Although the variance can be large due to the nature of LLM-based experiments, e.g., in Table 1, the current results show clear trends of relations between (1) CoT length vs. Acc., which can differ across tasks and models (Figure 1); (2) Sentiment vs. Acc (neutral sentiment outputs are likely to get higher accuracy); and (3) Lexical hints, e.g., hedging, and the accuracy.

However, most findings are drawn from a visualized presentation with accuracy as a function of the signals. To validate the main claim that these signals can serve as a calibration signal that complements “unreliable” self-reported probabilities (abstract), there should be numerical results showing that self-reported probs are not reliable and the signals studied in this work are (1) more reliable; (2) can assist self-reported probs. In this case, the calibration error should be shown to reveal the relation between accuracy and these signal sources (at instance/bucket level), i.e., besides the visualization, the authors should treat each / compositional signal (after normalization/rescaling) as the “confidence” to compute the calibration error against the per-example correctness. The current calibration errors shown in Table 1 are only on the performance of the self-reported accuracy – the proposed signals are not shown.

As a short summary, the reviewers and AE acknowledge (1) the effort of the authors during the review period and (2) the potential usefulness of the signals proposed and studied on the characteristics of the model CoT. The reviewer would suggest further improvement with (1) more rigorous analysis and numerical metrics to validate the effectiveness; (2) going deeper into slice-wise comparison on the good/bad traces within or across models.

**Resubmission Of Major Revision:**

The authors may consider submitting a major revision at a later time.